# Electron transport and visible light absorption in a plasmonic photocatalyst based on strontium niobate

D.Y. Wan[1,2], Y.L. Zhao[1], Y. Cai[3], T.C. Asmara[4], Z. Huang[1], J.Q. Chen[1], J. Hong[5], S.M. Yin[5], C.T. Nelson[6], M.R. Motapothula[1], B.X. Yan[1,2], D. Xiang[7], X. Chi[2,4], H. Zheng[3,6], W. Chen[2,7], R. Xu[5], Ariando[1,2], A. Rusydi[1,2,4], A.M. Minor[3,6], M.B.H. Breese[1,2,4], M. Sherburne[3], M. Asta[3], Q.-H. Xu[1,7] & T. Venkatesan[1,2,8,9,10]

Semiconductor compounds are widely used for photocatalytic hydrogen production applications, where photogenerated electron–hole pairs are exploited to induce catalysis. Recently, powders of a metallic oxide ($Sr_{1-x}NbO_3$, $0.03 < x < 0.20$) were reported to show competitive photocatalytic efficiencies under visible light, which was attributed to interband absorption. This discovery expanded the range of materials available for optimized performance as photocatalysts. Here we study epitaxial thin films of $SrNbO_{3+\delta}$ and find that their bandgaps are $\sim 4.1$ eV. Surprisingly, the carrier density of the conducting phase exceeds $10^{22}$ cm$^{-3}$ and the carrier mobility is only 2.47 cm$^2$ V$^{-1}$ s$^{-1}$. Contrary to earlier reports, the visible light absorption at 1.8 eV ($\sim 688$ nm) is due to the plasmon resonance, arising from the large carrier density. We propose that the hot electron and hole carriers excited via Landau damping (during the plasmon decay) are responsible for the photocatalytic property of this material under visible light irradiation.

[1] NUSNNI-NanoCore, National University of Singapore, Singapore 117411, Singapore. [2] Department of Physics, National University of Singapore, Singapore 117551, Singapore. [3] Department of Materials Science and Engineering, University of California, Berkeley, California 94720, USA. [4] Singapore Synchrotron Light Source, National University of Singapore, Singapore 117603, Singapore. [5] School of Chemical and Biomedical Engineering, Nanyang Technological University, Singapore 637459, Singapore. [6] Materials Science Division, Lawrence Berkeley National Laboratory, Berkeley, California 94720, USA. [7] Department of Chemistry, National University of Singapore, Singapore 117543, Singapore. [8] NUS Graduate School for Integrative Sciences and Engineering, National University of Singapore, Singapore 117456, Singapore. [9] Department of Material Science and Engineering, National University of Singapore, Singapore 117575, Singapore. [10] Department of Electrical and Computer Engineering, National University of Singapore, Singapore 117583, Singapore. Correspondence and requests for materials should be addressed to Y.L.Z. (email: yongliangrosy@gmail.com) or to T.V. (email: Venky@nus.edu.sg).

Converting solar energy into chemical energy (for example, splitting water by sunlight) with the aid of photocatalysts is a promising way to reduce the increasing demand for fossil fuels[1–10]. Very few oxide semiconductors have been used as photocatalysts, as they need to be chemically robust and their bandgap should be neither too wide nor narrow to absorb sunlight in the visible range efficiently and also satisfy the minimum energy requirement (1.23 eV theoretically but > 1.9 eV experimentally) for splitting water into hydrogen and oxygen[11–13]. Large bandgap oxides (such as $TiO_2$) are used in photocatalytic water splitting either by reducing their optical bandgap to absorb visible light or incorporating visible light absorbers such as organic dyes, low bandgap quantum absorbers or metal nanostructures[14,15] in the host. In the former case, cationic or anionic doping, or a combination of both, is typically applied to narrow the bandgap[16–19]. Unfortunately, in most cases, the bandgap change achieved is small mainly due to the fact that the discrete defect states are normally very close to the band edges[20–22]. In the latter case, hot electrons in the visible light absorber inject into the conduction band of large bandgap oxides, which are subsequently used to reduce water to hydrogen gas. Among the visible light absorbers, metal (for example, Au and Ag) nanostructures are special as the hot electrons generated by the decay of visible light excitation of surface plasmon resonance can be injected into a large bandgap semiconductor such as $TiO_2$ (refs 23,24). However, the materials used for enhancing the photocatalytic activity by using surface plasmon resonance are mainly Au, Ag nanostructures, which are not low-cost materials.

Recently a red metallic oxide $Sr_{1-x}NbO_3$ $(0.03 < x < 0.20)$ (in the form of powders) was used in photocatalytic hydrogen evolution (see Supplementary Note 1 and Supplementary Fig. 12) and the authors proposed a special band structure, in which the electron–hole pairs come from the optical transition from the metallic conduction band (about 1.9 eV above the valence band) to a higher level unoccupied band[25,26]. In these reports, the visible light absorption was attributed to the electron's interband transitions and the electron–hole pair separation was attributed to the assumption of high carrier mobility, although only temperature-dependent conductivity was measured. As both charge carrier density and mobility contribute to the conductivity of a sample, simply assuming a large mobility for a highly conductive material may lead to a wrong conclusion. It has also been reported by Oka et al.[27] that the highest conducting $SrNbO_3$ thin film grown on $KTaO_3$ substrate can have a high carrier density of $10^{22}$ cm$^{-3}$ and relatively low mobility of 16 cm$^2$ V$^{-1}$ s$^{-1}$ at room temperature. Besides carrier density and mobility, understanding the static optical properties (absorption, refractive index and loss function) and dynamics of the carriers would be crucial for understanding the details of the photo-catalytic process. The optical bandgap obtained from Kubelka–Munk transformation of the reflectance spectrum of the powder form $Sr_{1-x}NbO_3$ is inaccurate, as it neglects the plasmonic absorption. The proper way to measure the optical bandgap is to obtain the complex dielectric function, by using Kramers–Kronig-transformed reflectivity or spectroscopic ellipsometry. Epitaxial thin films are required for measuring the intrinsic mobility, as the grain boundaries in the film are much less than in powders enabling accurate Hall measurement of carriers and hence mobility. Using thin films also allows us to measure the transmittance, reflectance and ellipsometry spectra accurately, which can give proper optical and plasmonic absorptions of this material. Furthermore, the dynamical process of the hot electron can be investigated by femtosecond time-resolved transient absorption (TA) spectroscopy.

Here we have prepared $SrNbO_{3+\delta}$ films by pulsed laser deposition (PLD) at various oxygen partial pressures on top of insulating $LaAlO_3$ substrates and compared their optical spectra, electronic transport and carrier dynamics properties. We found the optical bandgap to be around 4.1 eV, which was almost independent of the oxygen content, although the crystal structure changed from pseudo-tetragonal perovskite to orthorhombic with increasing oxygen partial pressure from $5 \times 10^{-6}$ Torr to $1 \times 10^{-4}$ Torr. The plasmon peak for the film grown at $5 \times 10^{-6}$ Torr was found at about 1.8 ev (688 nm). The high conductivity ($\sim 10^4$ S·cm$^{-1}$) of the sample prepared at low pressure is mainly contributed by the high charge carrier density ($\sim 10^{22}$ cm$^{-3}$) rather than its mobility (2.47 cm$^2$ V$^{-1}$ s$^{-1}$) at room temperature. Thus, we believe that the photocatalytic activity of $SrNbO_{3+\delta}$ under visible to near-infrared (NIR) irradiation is due to the hot electrons generated from the decay of the plasmon in $SrNbO_3$ instead of interband absorption transition. Thus, $SrNbO_{3+\delta}$ represents an extraordinary material system, which has a large bandgap of 4.1 eV but a degenerate conduction band with a large carrier density exceeding $10^{22}$ electrons per cm$^3$, which leads to strong useful plasmonic effects.

## Results

**Characterization of strontium niobate thin films.** The X-ray diffraction spectrum of the film deposited at oxygen partial pressure of $5 \times 10^{-6}$ Torr is shown in Supplementary Fig. 1. The $\theta - 2\theta$ scan indicates the film's lattice parameter along out-of-plane [001] direction as 4.10 Å. The full width at half maximum of the rocking curve is measured to be 0.71°, which is acceptable by considering the large lattice mismatch between the film and the substrate ($LaAlO_3$ forms in pseudocubic perovskite structure at room temperature with lattice constant 3.79 Å). The reciprocal space maps (RSMs) of ($-103$) and (103) [(0–13) and (013)] are symmetric with respect to the out-of-plane [001] axis, suggesting orthogonality of [001] and [100] ([001] and [010]) axes. Despite their relatively broad RSM spots, the in-plane parameters are obtained as 4.04 Å equally. Hence, the film forms in the tetragonal-like structure on $LaAlO_3$ substrate with a large strain near the interface. The strain effect can be clearly seen in the local high-resolution transmission electron microscopy image of the lattice (Supplementary Fig. 1c). The highlighted open burgess circuit indicates an a[100]$_p$-type edge dislocation core, where the extra plane on the $LaAlO_3$ side indicates a compressive misfit of the film. As the oxygen partial pressure increases to $1 \times 10^{-4}$ Torr, a small shift of the film peaks towards low angles can be observed in the $\theta - 2\theta$ scan and the full width at half maximum of the rocking curve increases (Supplementary Fig. 2). Structural changes from tetragonal to orthorhombic were observed in the local high-resolution transmission electron microscopy images (Supplementary Fig. 3). The orthorhombic structure is close to the reported structure of $Sr_2Nb_2O_7$ (bulk $a = 3.933$ Å, $b = 26.726$ Å, $c = 5.683$ Å), which has an equivalent tetragonal structure with lattice parameters $a = b = 3.901$ Å and $c = 3.933$ Å. The decreasing of the out-of-plane and in-plane lattice parameters with oxygen partial pressure was obtained from the electron diffraction pattern (Supplementary Fig. 3 and Supplementary Table 1). At the intermediate oxygen partial pressure, mixed structure exists in the film.

It was reported that Sr content strongly determines the crystal structure of the non-stoichiometric $SrNbO_3$ phase[28,29]. Here the elemental content of the films deposited at different oxygen partial pressures are precisely studied (Supplementary Fig. 4 and Supplementary Table 2). Within the detection limit of proton-induced X-ray emissions and Rutherford backscattering spectroscopy, the film deposited at $5 \times 10^{-6}$ Torr has Sr:Nb:O = 1:1:3. Films deposited at higher oxygen pressures show the same Sr/Nb ratio but increased oxygen content.

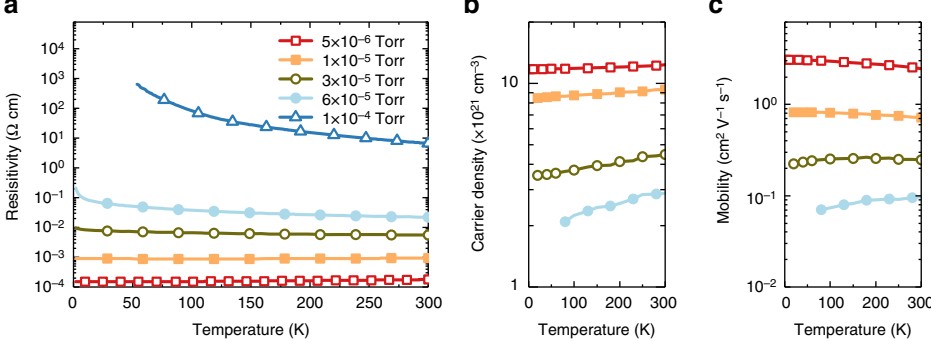

**Figure 1 | Electronic transport properties of SrNbO$_{3+\delta}$ films.** Temperature-dependent transport properties of the films prepared at various oxygen pressures: (**a**) resistivity, (**b**) mobile electron density obtained from Hall measurement and (**c**) electron mobility of the films.

**Electronic transport property of SrNbO$_{3+\delta}$.** The conducting property of the films show a transition from metallic to semiconductor transport behaviour when the deposition oxygen partial pressure is increased (Fig. 1a). The free electron density of the most conductive sample (prepared at $5 \times 10^{-6}$ Torr) reaches $10^{22}$ cm$^{-3}$, which agrees with the reported data of SrNbO$_3$ thin film grown on KTaO$_3$ (ref. 27) (Fig. 1b) indicative of a degenerate Fermi level. In contrast, the electron mobility is only 2.47 cm$^2$ V$^{-1}$ s$^{-1}$ at room temperature, which is not significant compared with other oxides (for example, TiO$_2$ and BaSnO$_3$) and semiconductors[30–33] (Fig. 1c). The room temperature mobility of our thin film is smaller than that of SrNbO$_3$ grown on KTaO$_3$ ($\sim 16$ cm$^2$ V$^{-1}$ s$^{-1}$) and this can be attributed to strain in the film as the lattice mismatch between SrNbO$_3$ and KTaO$_3$ (mismatch $= -0.85\%$) is smaller compared with that between SrNbO$_3$ and LaAlO$_3$ (ref. 27) (mismatch $= -5.77\%$). However, we can still conclude that the high conductivity of this material is mainly attributed to the high carrier density and not just the carrier mobility. Both the free electron density and mobility are almost independent of the measurement temperature, which indicates that the electronic transport property of SrNbO$_3$ is dominated by the free electrons in the conduction band and electron–electron scattering. The large free electron density and normal electron mobility of strontium niobate at room temperature imply the absence of significant internal electric field to avoid electron–hole recombination and the interband transition model proposed in the previous report[34] is not suitable. As the oxygen content increases in the film, the sample becomes more insulating. Both the charge carrier density and the mobility decreases with oxygen partial pressure, which is consistent with the observed two crystal structures of the materials[35]. Metallic SrNbO$_3$ forms in tetragonal perovskite structure at $5 \times 10^{-6}$ Torr and almost insulating SrNbO$_{3.5}$ forms in the orthorhombic structure at $1 \times 10^{-4}$ Torr. At the intermediate pressure, the semiconductor forms in cermet structure. The TEM images show excess oxygen forming sheet-like defect structures leaving regions with 113 stoichiometry in the midst (Supplementary Fig. 3). These insulating oxygen sheets create a cermet-like structure with increasing oxygen uptake.

The photocatalytic hydrogen evolution performance is discussed in Supplementary Note 1 and shown in Supplementary Fig. 12 as the hydrogen evolution efficiency is not the focus of this work.

**Ultraviolet–visible–NIR spectra of SrNbO$_{3+\delta}$.** Figure 2a shows the ultraviolet–visible–NIR transmission spectra of SrNbO$_{3+\delta}$ thin films deposited at various oxygen pressure. A cutoff of the transmission edge is observed near 300 nm (Fig. 2a), which

indicates an optical bandgap of $\sim 4.1$ eV (from Tauc plot indirect, Supplementary Fig. 5) and it is almost independent of the preparation oxygen pressure. The transmission above 600 nm of the films continuously increases with oxygen partial pressure above 600 nm, which indicates absorption along with free carrier absorption in this wavelength range, where the latter is consistent with the metallic nature of the films and powders[25,36]. Both the transmission and reflection of the film prepared at $5 \times 10^{-6}$ Torr were plotted, from which the accurate pure absorption spectrum could be obtained (Fig. 2b). As we can see in the reflection spectrum, the reflection above 700 nm is much larger than that below 600 nm, which is very similar to the reflection behaviour of plasmonic materials. The minimum reflection is located at around 600 nm, which could indicate the rough frequency of its plasmon resonance. Hence, the reflection spectrum between 500 and 1,000 nm can be well fitted by the Drude–Lorentz Model and the corresponding plasmon frequency of the fitting curve is 1.65 eV (750 nm). The plasmon resonance in the strontium niobate is a result of its extremely large free electron density as measured in its electronic transport property. It should be noted that there is only one obvious absorption peak near 675 nm at the visible range in the absorbance spectrum.

**Loss function of SrNbO$_3$ and its plasmon energy position.** The complex refractive index, $\tilde{n}(\omega) = n(\omega) + i\kappa(\omega)$, and the loss function, $-\text{Im}[\varepsilon^{-1}(\omega)]$, spectra of the $5 \times 10^{-6}$ Torr sample extracted from spectroscopic ellipsometry data are shown in Fig. 2c,d (for further details, see Supplementary Note 2, Supplementary Figs 6–8 and Supplementary Table 3). The extinction coefficient spectrum, $\kappa(\omega)$, of the sample (Fig. 2d) shows that it has a Drude peak below 2 eV (typical of a metal) and a first interband transition peak (indicating the bandgap of the film) above 4.1 eV, consistent with its transmission spectrum (Fig. 2b). Between these two peaks, the $\kappa(\omega)$ is featureless, indicating the lack of major optical transitions within the 2–4.1 eV energy range. Meanwhile, the loss function spectrum of the sample (Fig. 2d) shows a large peak at $\sim 1.5$–2.1 eV with a peak position of $\sim 1.8$ eV (688 nm), indicating the energy position of the plasmon resonance[37]. The energy position of plasmon obtained from the ellipsometry spectroscopy is more accurate compared with that obtained by the fitting of ultraviolet–visible–NIR reflection spectrum. From $n(\omega)$ and $\kappa(\omega)$ spectra, the normal-incident reflectivity of the film can be obtained using Fresnel equations, as shown in Fig. 2e. This reflectivity is consistent with the spectrum measured by ultraviolet–visible spectroscopy. From the reflectivity, the Kubelka–Munk function of the film can be obtained (Fig. 2e) and the shape of the function resembles the previous reported results, with an apparent

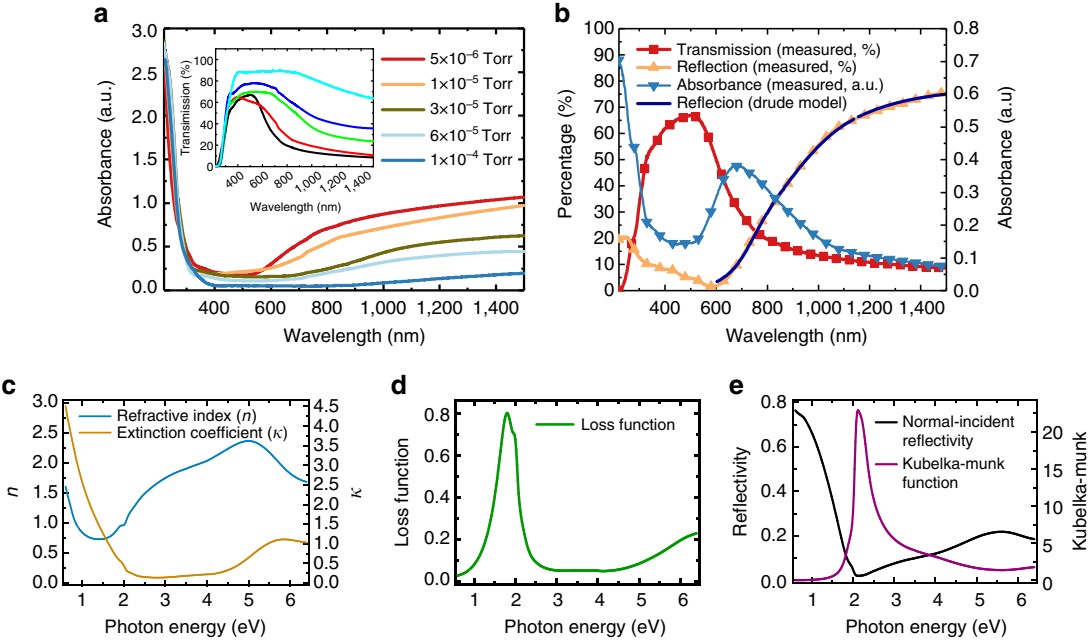

**Figure 2 | The optical properties of SrNbO$_{3+\delta}$ films.** (**a**) Ultraviolet–visible–NIR spectra of SrNbO$_{3+\delta}$ thin films deposited at various oxygen partial pressures, an absorption edge located at the wavelength of 300 nm can be observed. (**b**) The transmission, reflection spectra of the film deposited at $5 \times 10^{-6}$ Torr, with the absorbance spectrum obtained as plot in red. Ellipsometry analysis of (**c**) the refraction index ($n$) and extinction coefficient ($\kappa$). (**d**) Loss function and (**e**) reflectivity of SrNbO$_3$ as a function of photon energy.

absorption edge at $\sim 2\,\mathrm{eV}$. As there is no peak in the $\kappa(\omega)$ spectrum at around that energy, this absorption edge does not come from intra- or interband transition as previously reported. Instead, this absorption edge is plasmonic in origin, because it coincides with the plasmon peak at $\sim 1.8\,\mathrm{eV}$ in the loss function spectrum. The plasmon observed in ellipsometry spectra may be the surface Plasmon, because a similar peak near 675 nm can be observed in the ultraviolet–visible–NIR absorption spectrum and in general, the volume plasmon is not possible to be directly coupled with the photon[38,39]. As the photocatalytic hydrogen evolution experiments were performed with powders, coupling of the photons to the plasmons is not an issue. As the calculated penetration depth of the plasmon is 80 nm, which is comparable to the particle size ($\sim 100$–$1{,}000\,\mathrm{nm}$), further experiments are needed to distinguish between surface versus bulk (Supplementary Fig. 9). Hereon, we will refer to this as just plasmon.

**Band structure and density of states of SrNbO$_{3+\delta}$.** The energy band structures of SrNbO$_3$, SrNbO$_{3.4}$ and SrNbO$_{3.5}$ are calculated using density functional theory (DFT) and shown in Fig. 3. In the calculations, the perovskite structure is assumed for the stoichiometric SrNbO$_3$ compound and the extra oxygen atoms for the hyper stoichiometric compositions are assumed to order into planar defects, as illustrated by the structural figures in the left panels of Fig. 3. This structural model is consistent with electron microscopy data shown in Supplementary Fig. 3 (refs 40,41). The Fermi level of SrNbO$_3$ is in the conduction band, which implies that this material is metallic even though there is a bandgap as large as 3.0 eV between the CB and the highest fully occupied band (B$_{-1}$ band, reasonably close to our ultraviolet–visible spectrum considering that DFT calculations generally underestimate bandgap values). The Fermi level of SrNbO$_{3.4}$ is located near the bottom of the conduction band; thus, the conductivity is poorer than that of SrNbO$_3$, as there are fewer

states for the free carriers leading to a lower carrier density. These results are consistent with the experimental measurements. Unlike SrNbO$_3$ and SrNbO$_{3.4}$, the Fermi level of SrNbO$_{3.5}$ is located at the top of the valence band; thus, SrNbO$_{3.5}$ is insulating and the film should be transparent, consistent with experiments.

**Ultrafast carrier dynamics in SrNbO$_3$.** To further understand the role of the plasmon in the catalytic process, time-resolved pump–probe spectroscopy was used to characterize the carrier dynamics in SrNbO$_3$. Figure 4a shows the various excitation wavelength-dependent differential reflection (DR, $\Delta R/R$) spectra with various time delay times. Two peaks located near 600 nm (positive) and 670 nm (negative) are observed in the DR spectra. It should be noted that the sign of DR signal would usually be opposite to the sign of differential transmission signal (Supplementary Fig. 10)[42,43]. Usually, the TA could be positive or negative because the pump pulse induces population or depopulation of particular states, which can decrease (photobleaching) or increase (photoinduced absorption) the absorption of the probe pulse[43]. However, this principle may not be applied to our DR spectra. As SrNbO$_3$ is a metallic oxide, the heat capacity of electrons ($C_e$) is much smaller than that of the lattice ($C_L$), for example, $C_e \ll C_L$ and a short laser pulse can selectively heat the electrons as has been seen for gold nanoparticles[44]. It has been shown in this case that when the temperature of electrons increases, the intensity of plasmon band will decrease and its linewidth will increase, leading to bleaching of the plasmon band at the resonance and TA at the wings of the bleached spectrum[45]. Therefore, the negative peak near 670 nm can be attributed to the decrease (bleaching) of the plasmon band intensity as the 670 nm peak position is consistent with the plasmonic resonance peak measured by spectroscopic ellipsometry. The peak near 600 nm is the TA at the wings of the plasmon (derivative peak) due to the plasmon broadening

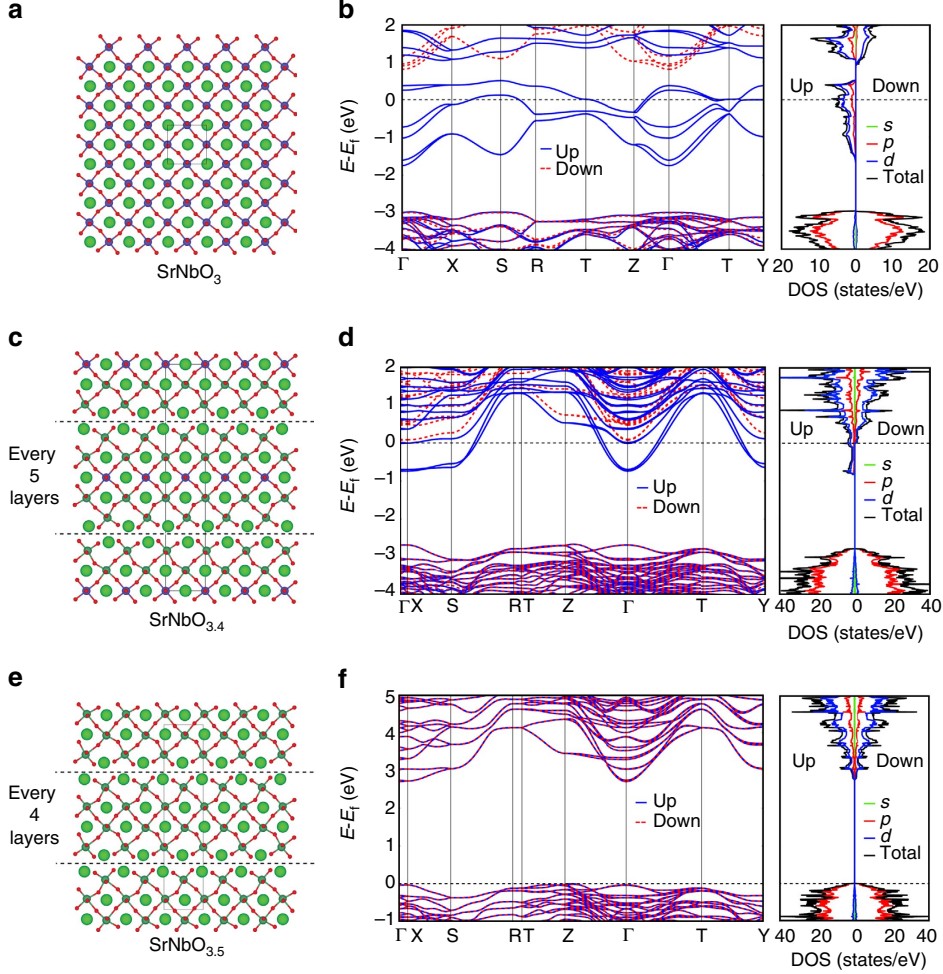

**Figure 3 | Crystal structures and energy band structures with its density of state (DOS) of SrNbO$_{(3+\delta)}$.** (a) Distorted perovskite structure of SrNbO$_3$. O$^{2-}$: small red sphere, Sr$^{2+}$: large green sphere, Nb$^{4+}$: small blue sphere. The unit cell is shown with the black solid line. (b) Band structure and DOS of SrNbO$_3$, showing its metallic behaviour. (c) The layered structure of SrNbO$_{3.4}$ with extra oxygen layers inserted every five octahedral layers. The dash lines indicate where the extra oxygen layers are. Nb$^{4+}$ (small blue sphere) and Nb$^{5+}$ (small green sphere) are given different colours to show charge ordering in this composition. (d) Band structure and DOS of SrNbO$_{3.4}$, with significantly reduced carriers at the Fermi level. (e) The layered structure of SrNbO$_{3.5}$, with an extra oxygen layer inserted every four octahedral layers. (f) Band structure and DOS of SrNbO$_{3.5}$, showing its insulating behaviour. The calculation method is using DFT + U (U = 4 eV).

induced by increased temperature electrons (see Supplementary Note 3 and Supplementary Figs 10 and 11 for a more detailed discussion of the assignment). The other wing at the lower energies is not seen due to the limitation of our spectrometer.

After the conduction electrons are excited to the unoccupied states by the pump pulse, the electrons will form a transient broad non-thermalized (non-Fermi) distribution above the Fermi level[44,46,47]. These hot electrons have a large excess energy above the Fermi level, much larger than thermal excitations at ambient temperatures[48,49]. The hot electrons will exchange their energy with the lower-energy electrons via electron–electron scattering process and form a Fermi distribution characterized by the electron temperature ($T_e$). The timescale of this process was reported to be around 100 fs to 1 ps[50,51]. The dynamics of the hot electrons within the first 2.0 ps in SrNbO$_3$ are shown in Supplementary Fig. 11. The excitation wavelength-dependent lifetimes of the hot electrons are around 120–250 fs by fitting them with a single exponential function. This is consistent with the reported timescale. However, it should be noted that the interaction between the pump pulse and probe pulse may still exist and affect the measurement results, as this lifetime is still on the order of the laser pulse duration ($\sim$150 fs). At this stage, the electron temperature ($T_e$) is still much

larger than the lattice temperature ($T_L$), for example, $T_e \gg T_L$. As the velocity of the hot electron is reduced, the interaction between electrons and lattice would increase via the electron-phonon coupling[44]. Thus, the lattice would be heated by the electrons and the phonons would be excited, which would take several picoseconds. At the final step, the remaining heat energy in the lattice would be lost to the surroundings through the phonon relaxation processes. This would take over a long timescale from 100 ps to 1 ns depending on the material[48]. Figure 4b shows the dynamics of the hot electrons with a much larger time range of about 1 ns. These dynamic curves could be well fitted by the biexponential function. The lifetimes of fast components are around 5–27 ps as shown in Fig. 4c and we assign this fast component to the process of the electron–phonon coupling. The slow components have a relatively longer lifetime ranging from 200 to 500 ps as shown in Fig. 4d, which should be attributed to the phonon relaxation process.

## Discussion

We have discussed that the optical absorption and electronic transport properties have proved that the plasmon exists and it

can be excited near 670 nm when under light irradiation in SrNbO₃. Based on these experimental and theoretical results, we propose that the photocatalytic property of SrNbO₃ can be attributed to the chemical property of the plasmon-induced hot electron–hole pairs. The expected working mechanism for $H_2$ evolution from aqueous oxalic acid solution over Pt loaded SrNbO₃ sample is shown in Fig. 5. When the strontium niobate is under visible light irradiation, the plasmon would be excited near

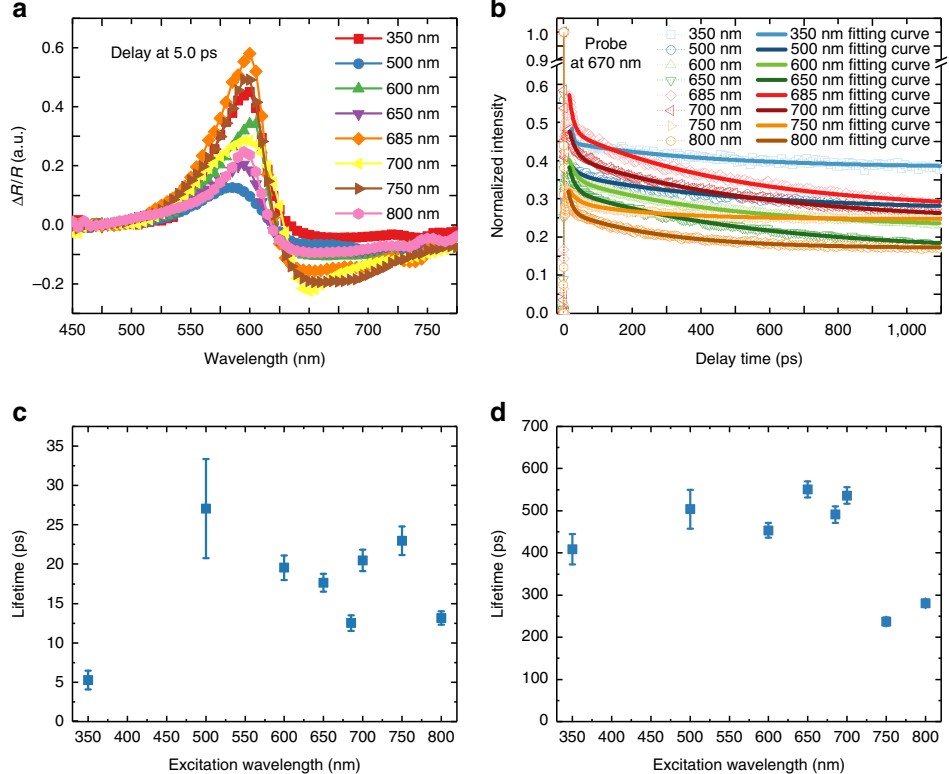

**Figure 4 | The TA spectra of SrNbO₃ thin films.** (**a**) The differential reflectance ($\Delta R/R$) spectra for the SrNbO₃ film of the delay time at delays of 5.0 ps with pump light of various wavelengths and a white light continuum probe. (**b**) The excitation wavelength-dependent dynamic spectra with the probe wavelength at 670 nm with a measurement range of 1,100 ps. The excitation wavelength-dependent carrier lifetimes are shown for two processes with the probe pulse at 670 nm: (**c**) the fast process corresponding to the electron–electron scattering and (**d**) the slow process corresponding to thermal dissipation.

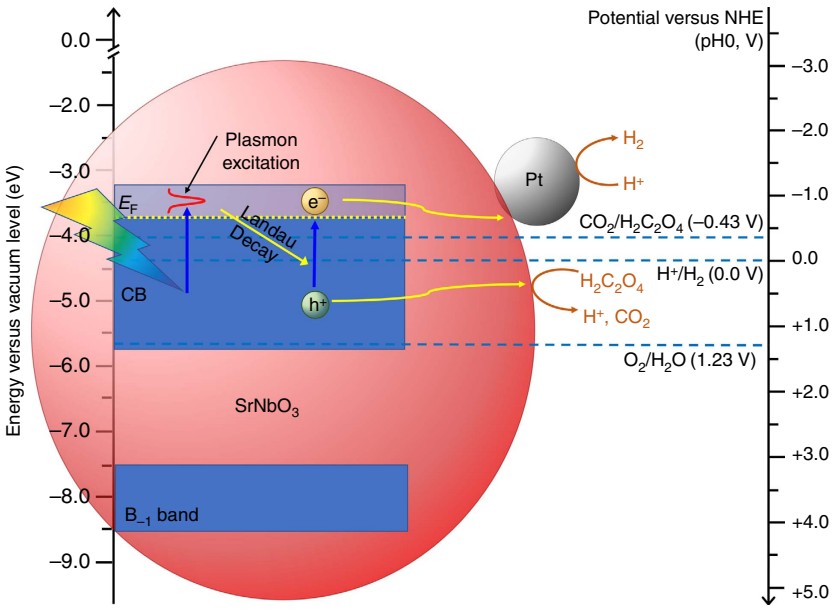

**Figure 5 | Schematic of the photocatalytic hydrogen evolution reaction on Pt-loaded SrNbO₃.** CB, conduction band; B₋₁ band, the highest fully occupied band below CB. The Fermi level (from UPS measurement) and the widths of CB and B₋₁ band shown (from DFT calculations) in this figure are for illustration purpose.

670 nm. The plasmon resonance can be damped by the generation of hot electron–hole pairs via Landau damping, which is a pure quantum mechanical process[48,52]. In Landau damping, a plasmon quantum can be transferred into a single electron–hole pair excitation on a timescale ranging from one to few hundreds of femtoseconds. The electrical field enhanced by the plasmon can also induce transitions of the conduction electrons from the occupied states (initial states below the Fermi level) to the unoccupied states (final states above the Fermi level[49]. The position of the Fermi level has been measured using ultraviolet photoelectron spectroscopy (UPS) (Supplementary Fig. 13) and it is 3.70 eV below the vacuum level). The holes will be left below the Fermi level in the conduction band. A transient non-Fermi hot electron distribution will form with a timescale $<100\,\mathrm{fs}$[48,51]. We assume that the hot carriers are mainly generated via the intraband transitions induced by plasmon decay, because the interband transition from CB to $B_1$ is not allowed, as it is a $d$ band to $d$ band transition. One part of the hot electrons will redistribute their energies among many lower energy electrons via electron–electron scattering at a timescale of a few picoseconds and then cool down by electron–phonon coupling and thermal dissipation[48,53]. The other part of the hot electrons will be injected into the Pt co-catalyst, which is an efficient electron trap centre and acts as the active site for the reduction reaction[54,55]. Thus, these hot electrons injected into Pt would reduce $H^+$ to $H_2$. As the energy distribution of the CB is very wide, the holes left in the electron-deficient $SrNbO_3$ particle would have enough energy to oxidize the oxalic acid to $CO_2$.

In summary, we have demonstrated that epitaxial, single crystal $SrNbO_3$ film can be obtained by PLD. The electron's mobility of this material is very normal, only $2.47\,\mathrm{cm^2\,V^{-1}\,s^{-1}}$ at room temperature, and there are no band absorptions within the gap ($\sim 4\,\mathrm{eV}$); thus, the interband transition model cannot be applied to explain the photocatalytic activity of this material. Further, the material has a degenerate conduction band with a gap of 4.1 eV but the large carrier density leads to a large plasmon at 1.6–1.8 eV, which simulates a mid-gap absorption. The hot electron–hole pairs can be generated in the conduction band of $SrNbO_3$ via the Landau damping during the decay of plasmons. Therefore, we propose that the sunlight is absorbed by the plasmons whose decay leads to hot carriers responsible for the catalytic reaction. Understanding of this first plasmonic metallic oxide and its use as a photocatalyst will open the doors for the design of a new family of photocatalytic materials.

## Methods

**Material preparation.** The PLD target was prepared by solid reactions of the $Sr_4Nb_2O_9$ precursor, Nb (Alfa Aesar, 99.99%, $-325$ meshes) and $Nb_2O_5$ (Alfa Aesar, 99.9985%, metals basis) powder mixtures in the proper molar ratio. The precursor was prepared by calcining $SrCO_3$ (Alfa Aesar, $>99.99\%$, metals basis) and $Nb_2O_5$ powder mixtures in a molar ratio of 4:1. The calcination and sintering were done in air and Ar gas environment for 20 h at a temperature of 1,200 and 1,400 °C, respectively. The films of $SrNbO_3$ were deposited on $LaAlO_3$ (001) substrates from these targets by PLD where a Lambda Physik Excimer KrF ultraviolet laser with the wavelength of 248 nm was used. The films were deposited at 750 °C, laser energy density of $2\,\mathrm{J\,cm^{-2}}$, laser frequency of 5 Hz and oxygen partial pressure of $5\times10^{-6}$ Torr to $1\times10^{-6}$ Torr. Typically, 130 nm-thick film could be obtained with half an hour deposition.

**Physical characterization.** The elemental compositions of the films were studied by 2 MeV proton-induced X-ray emissions with Si (Li) detector. The 3.05 MeV helium ion beam is used for oxygen resonance Rutherford backscattering spectroscopy measurements to enhance the sensitivity of backscattering signal from oxygen element. SIMNRA[56] simulation software is used to obtain the composition and thickness of the films. The obtained spectra were precisely fitted by SIMNRA simulation software. The crystal structure of the film was studied by high resolution X-ray diffraction (Bruker D8 with Cu $K\alpha1$ radiation, $\lambda = 1.5406$ Å) together with the RSMs. The local structure was measured by high-resolution scanning transmission electron microscopy (FEI Titan (Team0.5)@300 kV).

Optical bandgap of the film was measured by a ultraviolet–visible spectrophotometer (Shimadzu SolidSpec-3700). The transmissions of the films were measured and the corresponding absorption coefficients at particular wavelengths were derived from Beer–Lambert Law. Physical Properties Measurement System (Quantum Design, Inc.) was used to measure the electronic transport properties.

**Photocatalytic $H_2$ production measurement.** Light source used in the present experiment is 300 W Xe lamp with 410 nm long-pass filter. $H_2$ evolution was measured by suspending 50 mg sample powders together with 1 wt.% Pt co-catalyst in 100 ml oxalic acid aqueous solution (0.025 M). The evolved gasses were collected and quantified by an online gas chromatograph (Agilent 6890N, Argon as a carrier gas, 5 Å molecular sieve column and thermal conductivity detector (TCD)).

**TA measurement.** The electronic band structure of the films (all prepared at $5\times10^{-6}$ Torr, on $LaAlO_3$ substrate) and the lifetimes of the photo exactions were investigated by femtosecond time-resolved TA spectroscopy. The laser pulses were generated using a mode-locked Ti:sapphire oscillator seeded regenerative amplifier with a pulse energy of 2 mJ at 800 nm and a repetition rate of 1 kHz. The laser beam was split into two portions. The larger portion of the beam passed through a Light Conversion TOPAS-C optical parametric amplifier to generate 350 nm as the pump beam. The intensity of the pump beam was attenuated using a neutral density filter and modulated using an optical chopper at a frequency of 500 Hz. The smaller portion of the beam was used to generate white light by passing through a 1 mm sapphire plate, which acted as the probe beam. The white light beam was further split into two portions: one was used as the probe and the other was used as the reference to correct for the pulse-to-pulse intensity fluctuation. The pump beam was focused onto the sample surface with a beam size of 300 µm and it fully covered the smaller probe beam (diameter: 100 µm). The reflection of the probe beam from the sample surface was collected with a pair of lens and focused into a spectrometer. Very thick film samples were used to minimize the signal contribution from the substrate. The delay between the pump and the probe pulses was controlled by a computer-controlled translation stage (Newport, ESP 300). Pump–probe experiments were carried out at room temperature. During the measurements, the pump and the probe energies were kept low enough to minimize damage to the samples.

**Theoretical calculations.** The atomic and electronic structure of $SrNbO_{3+\delta}$ compounds were performed employing spin-polarized DFT calculations, using the Perdew–Burke–Ernzerhof[57] exchange-correlation potential, and the projector-augment wave method[58,59], as implemented in the Vienna *ab-initio* simulation programme[60]. In these calculations, Sr $4s4p5s$, Nb $4p5s4d$ and O $2s2p$ orbitals were treated as valence states, employing the projector-augment wave potentials labelled Sr_sv, Nb_pv and O in the Vienna *ab-initio* simulation programme Perdew–Burke–Ernzerhof library. The cutoff energy for the plane-wave basis set was set to 450 eV and the DFT + U approach due to Dudarev et al.[61] was employed to treat the Nb $4d$ orbitals occupied by the $Nb^{4+}$ ions present for $SrNbO_3$ and $SrNbO_{3.4}$ compositions with the value of $U$–$J$ set to 4 eV. $SrNbO_3$, $SrNbO_{3.4}$ and $SrNbO_{3.5}$ were modelled by supercells containing, respectively, 20 atoms with space group Pnam, 54 atoms with space group Pnnm and 44 atoms with space group Cmc2. For the cells with oxygen excess (that is, $SrNbO_{3.4}$ and $SrNbO_{3.5}$), the extra oxygen ions were placed in planar defects as illustrated in Fig. 3. In the structural relaxations, we employed using 8*8*4, 1*4*6 and 1*4*6 $k$-point meshes, for $SrNbO_3$, $SrNbO_{3.4}$ and $SrNbO_{3.5}$, respectively. The density of states was calculated with 16*16*8, 2*8*12 and 2*8*12 $k$-point meshes for the same three structures, respectively. In systems with occupied Nb $4d$ orbitals (that is, $SrNbO_3$ and $SrNbO_{3.4}$), we employed ferromagnetic ordering of the local moments on the $Nb^{4+}$ ions.

**Data availability.** The data that support the findings of this study are available from the corresponding author on reasonable request.

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

## Acknowledgements

We appreciate Dr Kokkoris from National University of Athens, Greece, for the carbon ions Rutherford backscattering spectroscopy measurement. Financial support from the Singapore-Berkeley Research Initiative for Sustainable Energy (SinBeRISE) programme is gratefully acknowledged. The NUS researchers acknowledge Singapore National Research Foundation under its Competitive Research Funding 'Control of exotic quantum phenomena at strategic interfaces and surfaces for novel functionality by in-situ synchrotron radiation' (NRF-CRP8-2011-06) and 'Oxide Electronics on silicon beyond Moore' (NRF-CRP15-2015-01), MOE-AcRF Tier-2 (MOE2015-T2-1-099) and FRC.

## Author contributions

D.Y.W. and Y.L.Z. developed this project, designed the research strategy and analysed the results. Y.L.Z., D.Y.W. and B.X.Y. prepared the samples and measured the electronic

transport properties. D.Y.W. carried out the TA spectra measurements with instruction by J.Q.C. Y.C. performed the DFT calculations. T.C.A. carried out ellipsometry spectroscopy characterization. X.C. and T.C.A. discussed the ellipsometry spectra. Z.H. measured the RSMs. J.H. carried out the photocatalytic experiments. C.T.N. performed the TEM imaging. M.R.M. measured the Rutherford backscattering spectroscopy (RBS) spectra. D.X. conducted and measured the UPS spectra supervised by W.C. R.X. supervised the photocatalytic measurement. H.Z. analysed the TEM images. A. supervised the RSMs experiments. A.R. supervised the ellipsometry spectra measurements. A.M. supervised the TEM imaging of the films. M.B.H.B. supervised the RBS experiments. M.A. supervised the DFT calculations. Q.-H.X. supervised the TA spectra measurements. D.Y.W and Y.L.Z. prepared this manuscript. T.V. supervised and led the project. All authors contributed to the scientific discussion and manuscript revisions.

## Additional information

**Competing interests:** The authors declare no competing financial interests.

**Publisher's note**: 

