## [Peer Review File · Nature Communications]

Reviewers' comments:

Reviewer #1 (Remarks to the Author):

This manuscript clearly describes the origin of red color of nonstoichiometric SrNbO₃ family is due to the plasmon excitation of mid-gap states by spectroscopic methods using thin films formed on LaAlO₃ by PLD. The conclusion is different from the previous literatures (refs. 25 and 26), which originally reported some photocatalytic H₂ evolution activity of the materials. I consider the present work presents the evidence of the red color being due to plasmonic excitation using reasonable experimental methods.

On the other hand, the mechanism of photocatalytic reaction seems not to be so clear from the manuscript. Are the hot electrons generated in the same mid-gap states or in an upper band? Although the authors mentioned in page 9 as follows " , which might be the transition from the deep trapped states to the conduction band", it is difficult to understand the detail. It may be better to depict such transitions in a schematic figure. Where are the holes to react with a reducing reagent (oxalic acid in the previous papers)?

Does the band gap energy of 4.1 eV correspond to the transition from the valence band to the conduction band in Figure 4(b)? Why are the transition from the valence band to the empty states of mid-gap and/or from the mid-gap to the conduction band not observed?

As a whole, I consider this is a nice work to reveal the origin of red color of SrNbO₃ family. But I am not quite sure if it is suitable to be published in Nat. Commun., because the original activity of this material is considerably low even with a sacrificial reagent, and it seems not to be so promising as photocatalysts and photoelectrodes.

Reviewer #2 (Remarks to the Author):

This paper reports a detailed analysis of the electronic structure of SrNbO₃ films and the mechanism of visible light absorption. It is argued that bulk plasmons are responsible for visible light absorption and photoreactivity, whereas previous studies suggested band gap excitations. The proposed mechanism is reasonably justified based on the static optical and electrical measurements. This is an interesting reinterpretation of the original claims on this system. However, before publication can be recommended the transient absorption measurements as well as plasmon dephasing mechanism must be re-considered.

The quoted hot electron lifetimes are about three orders of magnitude longer than in coinage metals. This makes sense based on the low mobility, however suggests that the electrons and holes are generated through plasmon dephasing in localized electronic states. The mechanism of plasmon dephasing by Landau damping occurs through the excitation of single electron hole pairs. Based on the proposed electronic structure and previous reports on this material there are no electronic transitions with energies in the range of the plasmon (~1.8 eV) that would produce localized electrons and holes with long lifetimes. This is a major issue in the interpretation. The plasmon must dephase somehow to produce e/h pairs and this is not described.

Related to this, could it be postulated what distribution of energy with respect to the E_f is contained in the electrons and holes produced from plasmon dephasing.

Because plasmon excitation is a resonant event why would 350 nm and 532 nm excitation give identical hot electron lifetimes and transient absorption decay behavior? The light absorption mechanism should be different. Why do both these excitations excite the plasmon?

Why does absorption of light at 800 nm cause significant changes in the electronic structure of the film at higher energies (550-700nm)?

In general the TA experiments did not seem consistently analyzed in any detail and thus the conclusions were not convincing.

Reviewer #3 (Remarks to the Author):

This paper purports to assign visible light absorption (leading to photocatalysis) in SrNbO₃ to plasmon absorption, leading to hot electrons. This is an exciting prospect as it would suggest diverse opportunities to harness plasmons in metal oxides in new application spaces and to simultaneously offer a new solution for efficient photocatalysis. Unfortunately, the data in the manuscript is far from convincing and the results are poorly contextualized relative to the existing literature. I do not suggest publication of this manuscript. Some details are described below.

Absorption is deduced from transmission and reflection. Is there no diffuse reflection/scattering? Further, in the discussion of this optical data, the authors mention that the absorption was fit by a Drude model, while on Fig 2b it appears the reflection spectrum was fit. In addition, there was a recent study that claimed the NIR extinction in SrNbO₃ couldn't be fit by the Drude model because of electron correlation (Oka, D.; Hirose, Y.; Nakao, S.; Fukumura, T.; Hasegawa, T. *Physical Review B*. 2015). All of this should be reconciled. More generally, correlation effects are not mentioned at all in this paper despite their importance in understanding this class of materials.

On page 7, the authors refer to a surface plasmon in their material, while in the rest of the manuscript the visible light absorption is ascribed to a volume plasmon.

The finding that the carrier concentration for SrNbO₃ is high, rather than mobility, is not new. This has been found in several other reports (Oka et al, *Phys Rev B*, 2015, Zhang et al, *Nature Mat*, 2015, etc.).

The authors should better explain why excitation of a plasmon (as opposed to interband excitation) would get around the problem of needing efficient electron-hole separation to carry out catalysis. Is the observed 200 ps decay time so much longer than expected relaxation time for an interband excitation? I don't think this is likely to be the case.

The interpretation of transient reflection spectra is unclear. The derivative lineshape suggests a peak shift while the authors interpret the positive and negative components of the peak to different aspects of the excited state electronic structure.

The transient data are also very limited - what about the dynamic evolution of the other peaks/troughs? This would help rationalize the assignment of the different spectral features.

This statement on page 9 requires a reference: "This structural model is consistent with electron microscopy analyses that will be reported elsewhere."

In the conclusion, the authors refer loosely to "The lifetime of the plasmon" which would usually be taken to mean the coherence time. I think they mean to interpret their time decay as the cooling time for hot electrons?

Perhaps the authors can clarify how the effects of varying in oxygen stoichiometry compares to the variation of the Sr/Nb ratio in this system, which is well studied in the literature.

It is unclear how the structures used for DFT relate to the experimental structures studied in this paper (not just generally in the literature). This needs to be better established to justify comparison of the DFT results to the experimental observations.

Reviewer #4 (Remarks to the Author):

The authors measure the transmittance, reflectance, and ellipsometry spectra of thin films of $\text{Sr}_{0.94}\text{NbO}_{3+\delta}$ in order to understand the relative effects of optical and plasmonic absorption. They show that the material has a 4.1 eV band gap and, for films grown at low oxygen partial pressure, an unusually large electron carrier density of around 10^{22} in the conduction band. For films grown at higher oxygen partial pressure, they find significantly lower charge carrier densities, with the films becoming insulating at the highest oxygen partial pressures. They also show that the mobility of the charge carriers is relatively low, even for the low oxygen partial pressure films. They therefore conclude that the high conductivity of these films is due to the high charge carrier density and not high mobility as suggested in previous work. Based on these results, they hypothesize that a large bulk plasmonic absorption at around 1.8 eV that generates hot electrons is responsible for the previously observed high photocatalytic activity of this material. They provide evidence to support this hypothesis by performing ellipsometry measurements to extract the complex refractive index and loss function. As the former show a large peak in the range 1.5-2.1 eV while the later has no peak in that energy range, they conclude that the mid-gap absorption is due to the bulk plasmon and not from an intra- or inter-band transition as previously reported. Finally, they show that the lifetime of the excited electrons is around 280ps when excited above the plasmon resonance, while it is only 63ps when excited below the plasmon resonance.

The authors perform a nice set of experiments demonstrating that the explanation for the high catalytic activity of this material given in previous work is incorrect, providing strong evidence that it instead arises from the large bulk plasmon which is due to the unusually large charge carrier density in this "wide band gap" material.

A key aspect of the paper that is missing, however, is a fundamental understanding of WHY the charge carrier density is so high in the first place. Many semiconducting complex oxides can be made with a wide variation in oxygen stoichiometry, yet do not exhibit this large carrier density. In many cases, this is due to the formation of compensating defects. What is special about this material that seemingly prevents compensating defects from trapping the excess electrons? The authors briefly mention that the degeneracy of the conduction band edge plays a role. However, although they include some rather superfluous DFT calculations, they do not even analyze the composition or degeneracy of the bands. Further, it would be very informative to investigate the energetics of some model compensating defects in order to understand the fundamental origins of the fascinating behavior of this material.

In sum, in order to warrant publication in Nature Communications, the authors need to at the very least include a more in-depth discussion of the physics that gives rise to the properties of the system.

Note: The reviewers' comments or questions are in black. Our response to them and modification made in manuscript are highlighted in blue and yellow, respectively.

Reviewers' comments:

Reviewer #1 (Remarks to the Author):

Comment 1.1. Are the hot electrons generated in the same mid-gap states or in an upper band? Although the authors mentioned in page 9 as follows " which might be the transition from the deep trapped states to the conduction band", it is difficult to understand the detail. It may be better to depict such transitions in a schematic figure.

Response 1.1:

The deep trapped states we observed by the transient absorption spectroscopy usually act as the recombination center for the photon generated electron-hole pairs. So the hot electrons cannot be generated from such deep trapped states. We have shown that there is a large plasmon resonance peak in the loss function spectrum (Fig. 2d), which also cause the large photon absorption near 1.8eV. Then the plasmon excitation can decay into hot electrons and holes.

Fig. R1 The schematic figure of the excitation mechanism. The work function of SNO is determined by UPS.

The Fig. R1 is the schematic figure of the excitation mechanism obtained from UPS measurements. As we can see in this figure, when SrNbO₃ is under irradiation, the free electrons will absorb the photon at the plasmon resonance (fig.2 (e) to fig. 2 (f)). When the plasmon decays hot electrons-holes are created.

Comment 1.2. Where are the holes to react with a reducing reagent (oxalic acid in the previous papers)?

Response 1.2:

In the photocatalytic efficiency measurement, we use Pt (1 wt.%) as the co-catalyst. Thus during the water splitting chemical reaction, the hot electrons generated in SrNbO₃ will be transferred from SrNbO₃ to Pt which will reduce the H⁺ to H₂. The holes left in the SrNbO₃ will oxidize the reducing reagent (oxalic acid).

Comment R1.3. Does the band gap energy of 4.1 eV correspond to the transition from the valence band to the conduction band in Figure 4(b)? Why are the transition from the valence band to the empty states of mid-gap and/or from the mid-gap to the conduction band not observed?

Response 1.3:

Yes, the band gap of 4.1eV correspond to the transition from the valence band to the conduction band. Our material SrNbO₃ is a metallic oxide, so the degenerate Fermi level is located in the conduction band, which has also been confirmed by DFT calculation (Fig. 4 (a)). The mid gap states have very low density and are related to deep trap states. These because of their low density do not show up either in the UV-Vis or the ellipsometry data. However, we can see them in the transient reflectivity data (Fig. 4 (a and b)) and these trap states act as recombination centers for the excited electron hole pairs (at 600 nm) and their sign is opposite to that of the of the resonant plasmon (at 700 nm).

Reviewer #2 (Remarks to the Author):

Comment 2.1: The quoted hot electron lifetimes are about three orders of magnitude longer than in coinage metals. This makes sense based on the low mobility, however suggests that the electrons and holes are generated through plasmon dephasing in localized electronic states. The mechanism of plasmon dephasing by Landau damping occurs through the excitation of single electron hole pairs. Based on the proposed electronic structure and previous reports on this material there are no electronic transitions with energies in the range of the plasmon (~1.8 eV) that would produce localized electrons and holes with long lifetimes. This is a major issue in the interpretation. The plasmon must dephase somehow to produce e/h pairs and this is not described.

Response 2.1:

We would like to thank the referee for bringing up this plasmon-assisted electron-hole pair creation issue. We admit that in the previous manuscript we did not discuss this in much details, and below we explain our analysis on how the plasmon decay can create the electron-hole pair necessary for the photocatalysis.

From Fig. 5b, it can be seen that the half-filled conduction band of SrNbO₃ at the Fermi level (labelled as CB) is composed of mainly Nb-4d states, which means that the metallic electrons are mostly d-electrons. At around 1-2 eV (and possibly even higher) above this conduction band, there is another completely unoccupied band (labelled as B1) that also mostly has Nb-4d characteristics.

Previous study (Nature Mater.) postulated that the electron-hole pairs necessary for the photocatalysis are created by direct photon excitations of metallic 4d electrons from CB to B1 band. However, our extinction coefficient spectrum (Fig. 2c) extracted from spectroscopic ellipsometry shows that there is no significant peak at ~2 eV, which is the supposed band-gap between CB and B1

bands. This means that the d-d transition between CB and B1 bands is not an optically-allowed transition, and photons cannot directly excite the metallic electrons and create the electron-hole pair.

This is where the loss function is crucial information. The loss function spectrum (Fig. 2d) shows that there is a large (volume) plasmon peak at ~ 1.8 eV, which indicates that the photon absorption is due to plasmon excitation instead of direct d-d interband transition. This plasmon excitation can then decay into electron-hole pair by Landau damping leading to hot electrons-holes

To conclude, based on our result, the process of photocatalysis is the following: photon \rightarrow plasmon \rightarrow plasmon decay by Landau damping \rightarrow e-h creation \rightarrow e-h separation \rightarrow photocatalysis.

Comment 2.2. Related to this, could it be postulated what distribution of energy with respect to the E_f is contained in the electrons and holes produced from plasmon dephasing.

Response 2.2:

Fig. 5b shows that the occupied part of CB spreads from 0 eV to ~ 2 eV below the Fermi level. Meanwhile, from the loss function spectrum (Fig. 2c), it can be seen that the plasmon peak is at ~ 1.8 eV with a FWHM of ~ 0.6 eV, which means that the plasmon has an energy distribution of $\sim 1.8 \pm 0.3$ eV. This means that the energy distribution of the created electron-hole pair has to conform with the plasmon energy distribution. This gives us a hole energy distribution of $0 - 0.8 \pm 0.3$ eV below the Fermi level and an electron energy distribution of $1 - 1.8 \pm 0.3$ eV above the Fermi level.

Comment 2.3. Because plasmon excitation is a resonant event why would 350 nm and 532 nm excitation give identical hot electron lifetimes and transient absorption decay behavior? The light absorption mechanism should be different. Why do both these excitations excite the plasmon?

Response 2.3:

We have remeasured all the transient dynamics data and have plotted the various lifetimes in Fig. 4 replacing the original data in Fig. 3. The difference here is that we have now made the measurement at a number of excitation wavelengths and have plotted the various decay times as a function of the pump wavelength. For the short time scale corresponding to Landau damping region we see the decay time peaking close to the plasmon resonance. The longer thermal decay time shows a step like behavior at the plasmon resonance showing clearly a difference in the plasmon behavior depending on the excitation wavelength. The quality of the data is superior to the prior data as the femtosecond laser system has been significantly improved over the last six months.

Comment 2.4. Why does absorption of light at 800 nm cause significant changes in the electronic structure of the film at higher energies (550-700nm)?

Response 2.4:

We would like to first note that in pump-probe experiments used to obtain the transient absorption, the "pump" part of the experiment is used to momentarily excite electrons from the ground state into excited states. Thus, the pumping photon would make some of the initially-occupied states to be momentarily unoccupied, and some of the initially-unoccupied states to be momentarily occupied. The "probe" part would then probe this excited state using another photon after a variable time delay. In the case where the pumping photon has a relatively low energy, it can still

affect the absorption spectra at higher energies because some of the momentarily-occupied (or -unoccupied) states might also be utilized by other, higher-energy transitions.

Comment 2.5. In general, the TA experiments did not seem consistently analysed in any detail and thus the conclusions were not convincing.

Response 2.5:

Please see the revised figure 4 and associated discussions in the paper.

Reviewer #3 (Remarks to the Author):

Comment 3.1. Absorption is deduced from transmission and reflection. Is there no diffuse reflection/scattering?

Response 3.1:

We would like to clarify that the optical experiments were performed on smooth thin film samples. The low surface roughness of the samples minimized diffuse reflections and scatterings, which allows the absorption to be directly deduced from transmission and reflection data.

Comment 3.2. Further, in the discussion of this optical data, the authors mention that the absorption was fit by a Drude model, while on Fig 2b it appears the reflection spectrum was fit.

Response 3.2:

We apologize for this mistake. The reflection spectrum was fit by Drude model, not the absorption spectrum.

Comment 3.3. In addition, there was a recent study that claimed the NIR extinction in SrNbO₃ couldn't be fit by the Drude model because of electron correlation (Oka, D.; Hirose, Y.; Nakao, S.; Fukumura, T.; Hasegawa, T. Physical Review B. 2015).

All of this should be reconciled. More generally, correlation effects are not mentioned at all in this paper despite their importance in understanding this class of materials.

Response 3.3:

We would like to clarify that in our analysis, we also could not fit the NIR part of our spectroscopic ellipsometry data using simple, isotropic Drude model (see supplementary for details). While we do not deny the possibility that correlation effects might play a role in the SrNbO₃, we found that the inability of simple Drude model to fit the NIR part of the spectra can be rectified more simply by considering the anisotropic nature of the film.

The XRD data (Fig. 1a) shows that the out-of-plane lattice constant of the film is 4.10 Å, while its in-plane lattice constants are 4.04 Å equally. This means that our SrNbO₃ film has a slight uniaxial anisotropy along the out-of-plane direction, which can affect the optical response of the film.

Indeed, as explained in the supplementary, we are not able to fit the spectroscopic ellipsometry data of the film using simple, isotropic Drude model (consistent with the PRB paper cited by the referee) due to this uniaxial anisotropy.

To rectify this, we instead fit the spectroscopic ellipsometry data using anisotropic Drude model, where the complex dielectric function of the film (including those inside the Drude region), $\epsilon = \epsilon_1 + i\epsilon_2$,

is taken to be different along the in-plane (i.e., ordinary) and out-of-plane (i.e., extraordinary) directions. With this anisotropic model, we are able to fit the spectroscopic ellipsometry data very well (Fig. S6). The fitted ordinary and extraordinary ϵ spectra of the film is shown in Fig. S7. It can be seen that although the features of both ordinary and extraordinary ϵ spectra of the film are generally quite similar, there are some differences with respect to relative peak heights, particularly in the Drude region below 2 eV. This indicates that the anisotropy indeed affects the optical response of the film, resulting in different Drude responses along different directions in the film.

Comment 3.4. On page 7, the authors refer to a surface plasmon in their material, while in the rest of the manuscript the visible light absorption is ascribed to a volume plasmon.

Response 3.4:

Yes, the visible light absorption is due to volume plasmon instead of surface plasmon. We have fixed this in the revised manuscript.

Comment 3.5. The finding that the carrier concentration for SrNbO₃ is high, rather than mobility, is not new. This has been found in several other reports (Oka et al, Phys Rev B, 2015, Zhang et al, Nature Mat, 2015, etc.).

The authors should better explain why excitation of a plasmon (as opposed to interband excitation) would get around the problem of needing efficient electron-hole separation to carry out catalysis. Is the observed 200 ps decay time so much longer than expected relaxation time for an interband excitation? I don't think this is likely to be the case.

Response 3.5:

First of all, we would like to clarify that the reason why we think that it is plasmon that drives the photocatalysis instead of direct interband transition is because we find no significant peak at ~2 eV in the extinction coefficient spectrum (Fig. 2c) and instead a large plasmon peak at ~1.8 eV in loss function spectrum is observed (Fig. 2d). This means that any interband transition around that energy (the Nb d-d transition according to the theoretical band structure in Fig. 4b) are not optically allowed. Instead, we find that around 1.8 eV, there exists a large (volume) plasmon in the loss function spectrum, which means that any light absorption around this energy should be attributed to plasmonic excitation instead of interband transition. This plasmonic excitation then might facilitate the optically-forbidden d-d transition, which creates the e-h pair necessary for the photocatalysis.

Therefore, in our interpretation, the photocatalysis process is the following: photon -> plasmon -> plasmon decay via Landau damping -> e-h creation -> e-h separation -> photocatalysis.

Comment 3.6. The interpretation of transient reflection spectra is unclear. The derivative lineshape suggests a peak shift while the authors interpret the positive and negative components of the peak to different aspects of the excited state electronic structure.

The transient data are also very limited - what about the dynamic evolution of the other peaks/trough? This would help rationalize the assignment of the different spectral features.

Response 3.6:

Our femtosecond transient absorption system has undergone significant improvements in the last six months. In the most recent data obtained which now has now replaced our original data there is less structural differences between the curves for different excitation wavelengths as shown in Figure 4 (a-b). The improved transient data (Figure 4 (c-f)) has now replaced the original data and it is clear that the decay times are dependent on the excitation wavelength as stated earlier.

Comment 3.7. This statement on page 9 requires a reference: "This structural model is consistent with electron microscopy analyses that will be reported elsewhere."

Response 3.7:

We apologize for this mistake. The electron microscopy data can be seen in:

1. Chen, C. et al. Atomic and electronic structure of the SrNbO₃/SrNbO_{3.4} interface. Appl. Phys. Lett. 105, 221602 (2014).
2. Lichtenberg, F., Herrnberger, A., Wiedenmann, K. & Mannhart, J. Synthesis of perovskite-related layered AnBnO_{3n+2} = ABO_x type niobates and titanates and study of their structural, electric and magnetic properties. Prog. Solid State Chem. 29, 1-70 (2001).

We have modified the above sentence.

Comment 3.8. In the conclusion, the authors refer loosely to "The lifetime of the plasmon" which would usually be taken to mean the coherence time. I think they mean to interpret their time decay as the cooling time for hot electrons?

Response 3.8:

Yes, in that sentence we meant to refer to the lifetime of the hot electrons instead of plasmon lifetime. We have fixed this in the revised manuscript.

The 400 ps lifetime refers to lifetime of the hot electrons, not plasmons. In fact, according to our estimation based on the FWHM of the plasmon peak, the plasmon lifetime, or more exactly, the plasmon dephasing/decoherence time, is in the 100s of femtosecond time scale expected for Landau damping. So, the plasmon decays into e-h pair after a few 100 fs, then the e-h pair lives for longer times to drive the photocatalysis.

Fig. R2 Typical time courses of H₂ evolution of strontium niobates powder under irradiation in the visible to NIR region, which is oxidized in air at various temperatures for 1 hour, in aqueous oxalic acid solution (0.025M) with 50mg catalyst.

Comment 3.9. Perhaps the authors can clarify how the effects of varying in oxygen stoichiometry compares to the variation of the Sr/Nb ratio in this system, which is well studied in the literature

Response 3.9:

Thanks for this suggestion. We have studied the effect of varying oxygen stoichiometry and show the results in our presented work. Compared with the variation of the Sr/Nb ratio, there will be a significant change in the crystal structure, electrical transport and optical absorption properties with the variation in oxygen stoichiometry. We have shown the TEM images of the SrNbO₃, SrNbO_{3.4} and SrNbO_{3.5} (Fig. S2). As we can see in Fig. S2, there is an extra oxygen layer in SrNbO_{3.4} and SrNbO_{3.5}, which is not observed in SrNbO₃. As we can see in the Fig. 3 (a) (b), the Resistivity of SrNbO_{3.4} and SrNbO_{3.5} as a function of temperature show the semiconductor behavior while the SrNbO₃ shows the metallic behavior. The Fig. R2 shows the effect of oxygen stoichiometry on the photocatalytic water splitting efficiencies. It is very interesting to find that the mixed-phase SrNbO_{3+x} shows the highest photocatalytic efficiencies compared with pure SrNbO₃ and SrNbO_{3.5}. The variation of Sr/Nb ratio in powder based catalysts show a maximum in activity at a ratio of about 0.9 close to our composition. Variation of this ratio in thin films will be a subject of future study and is beyond the scope of this work.

Comment 3.10: It is unclear how the structures used for DFT relate to the experimental structures studied in this paper (not just generally in the literature). This needs to be better established to justify comparison of the DFT results to the experimental observations.

Response 3.10:

The structures used in the DFT calculations are based on experimental crystal-structure refinements reported in the Inorganic Crystal Structure Database (ICSD) for SrNbO_3 , $\text{SrNbO}_{3.4}$ ($\text{Sr}_5\text{Nb}_5\text{O}_{17}$), and $\text{SrNbO}_{3.5}$ ($\text{Sr}_2\text{Nb}_2\text{O}_7$) [1-3]. Consistent with the HRTEM observations performed in the present work, the excess oxygen atoms (relative to the reference SrNbO_3 composition) in the $\text{SrNbO}_{3.4}$ and $\text{SrNbO}_{3.5}$ structures are accommodated in planar defects separating the corner-shared NbO_6 octahedra. As illustrated in Fig. 5, the average spacing of these planar defects becomes shorter the higher the oxygen to metal ratio.

Reviewer #4 (Remarks to the Author):

Comment 4.1. A key aspect of the paper that is missing, however, is a fundamental understanding of WHY the charge carrier density is so high in the first place. Many semiconducting complex oxides can be made with a wide variation in oxygen stoichiometry, yet do not exhibit this large carrier density. In many cases, this is due to the formation of compensating defects. What is special about this material that seemingly prevents compensating defects from trapping the excess electrons? The authors briefly mention that the degeneracy of the conduction band edge plays a role. However, although they include some rather superfluous DFT calculations, they do not even analyze the composition or degeneracy of the bands. Further, it would be very informative to investigate the energetics of some model compensating defects in order to understand the fundamental origins of the fascinating behavior of this material.

Response 4.1:

To understand the origin of the high carrier densities it is important to start by defining defects relative to the insulating compound $\text{SrNbO}_{3.5}$. This compound is a charge-transfer insulator in which the nominal oxidation state of Nb is 5+ (i.e., with $4d^0$ electron configuration). In this reference insulating state the valence band maximum is comprised mainly of oxygen 2p states, and the conduction band minimum of Nb-d states, as shown in panel f of Fig. 4. Relative to this reference insulating state, a reduction of oxygen to metal ratio in $\text{SrNbO}_{3+\delta}$ (with $\delta < 0.5$) can be considered as being accommodated through the formation of oxygen vacancies. These oxygen vacancies are donor-like and at high concentration they have the effect of pushing the Fermi level into the conduction band, leading to occupation of electronic states with predominant Nb-4d character, as shown in panels d and b of Fig. 5.

The referee is correct that compensating defects can form in this compound. For example, the role of Sr vacancies has been discussed in the literature previously (see, e.g., [4]). Sr vacancies are acceptors that can serve to compensate oxygen vacancies. It has been reported [4] that high Sr vacancy concentrations (at the level of 20 percent Sr site fraction) can lead to a reduction in the calculated band gap, in addition to shifting the Fermi level to lower values within the Nb-d conduction band in SrNbO_3 . However, in the present work the Sr vacancy concentration is measured to be much lower, at approximately 6 %. In the $\text{Sr}_{0.94}\text{NbO}_3$ samples considered in this work, the oxygen vacancies (i.e., the oxygen deficiency relative to the insulating composition $\text{SrNbO}_{3.5}$) are not fully compensated by the Sr vacancies, such that there results a high concentration of electron carriers derived from the conduction-band states with predominant Nb-4d character.

- [1] H. Hannerz, G. Svensson, S. Ya. Istomin, and O. G. D'yachenko, "Transmission Electron Microscopy and Neutron Powder Diffraction Studies of GdFeO₃ Type SrNbO₃," *J. Solid State Chem.* 147, 421-428 (1999)
- [2] S. C. Abrahams, H. W. Schmalke, T. Williams, A. Reller, F. Lichtenberg, D. Widmer, J. G. Bednorz, R. Spreiter, Ch. Bosshard and P. Gunter, "Centrosymmetric or Noncentrosymmetric? Case Study, Generalization and Structural Redetermination of Sr₅Nb₅O₁₇," *Acta Cryst.* B54, 399-416(1998)
- [3] P. Daniels, R. Tamazyan, D. A. Kuntscher, M. Dressel, F. Lichtenberg and S. van Smaalen, "The incommensurate modulation of the structure of Sr₂Nb₂O₇," *Acta Cryst.* B58, 970-976 (2002)
- [4] C. Sun and D. J. Searles, "Electronics, Vacancies, Optical Properties, and Band Engineering of Red Photocatalyst SrNbO₃: A Computational Investigation," *J. Phys. Chem. C* 118, 11267-11270 (2014).

Reviewers' comments:

Reviewer #1 (Remarks to the Author):

This is the revised manuscript of the previous one including some new experimental results and an attempt to explain the mechanism of a photocatalytic reaction.

Although I have tried carefully to understand the essential points of this work, there still exist several serious questions about the interpretations of the results and the mechanism of photocatalytic reactions as follows:

1) Fig. S6 is the band diagram of the SNO that was presented by the authors. At first, the sign of the vertical axis (potential vs. NHE) should be reversed. Then, l. 153-162 in the text, the authors mentioned the holes of this material drive the oxidation reaction, i.e. the oxidation reaction of oxalic acid (not water oxidation reaction). Where is the redox potential (or the oxidation potential of oxalic acid) in Fig. S6?

2) From l. 179 to l. 211 in the text, the authors attempted to explain the mechanism of photocatalytic reaction based on the results of time resolved pump-probe measurements. In the explanation of the results, the authors suddenly (?) introduced "deep trapped states", which confused me to understand the mechanism. Actually, where are the deep trapped states in the diagram of Fig. S6? In addition, I afraid the authors made a wrong interpretation about Fig. 4(d). The authors mentioned that plasmon resonance increases the lifetime of hot carriers because the lifetime of the transition from deep trapped states to the conduction band plotted in Fig. 4(d) drastically increases when the energy of pump pulse is higher than that of the plasmon resonance (Lines 198-211). However, the authors also mentioned that the energy of transition from deep trapped states to conduction band and that of plasmonic resonance is approximately same (Lines 188-191). Therefore, the pump pulse with energy lower than plasmon resonance cannot excite an electron in deep trapped states to conduction band and the transient absorption spectroscopy in this energy region might probe other minor processes.

3) From l. 212 to l. 225, the authors discussed band gap structures of three different compositions by DFT calculations in Fig. 5, i.e. SrNbO₃ (b), SrNbO_{3.4} (d) and SrNbO_{3.5} (f). To the reviewer's understanding from the text, the authors regarded SrNbO₃ (b) being corresponded to the sample prepared at the lowest O₂ pressure (5x10⁻⁶ Torr), which was mainly discussed in the present manuscript. If it is correct, the band structure is the same one which was reported in the previous work of Nat. Mater. in 2012. Of course, it is quite reasonable to interpret the red color of this material by plasmon resonance as proved by the former half of the present work. I believe that part is a nice piece of work, but it is not reasonably connected to the photocatalytic reaction mechanism.

4) In Fig. S5, the authors made an effort to show the photocatalytic activity of the SNB. Unfortunately, the activity of hydrogen evolution is very low in an aqueous solution of oxalic acid. The authors should estimate the quantum efficiency of the reaction. In addition, what will be the maximum quantum efficiency of hot electron transfer to Pt from plasmon excitation, which would be an interesting question?

5) It is better to cite and to discuss the literature below:

Oka, D.; Hirose, Y.; Nakao, S.; Fukumura, T.; Hasegawa, T. Phys. Rev. B 2015, 92, 205102.

As a whole, I agree the finding of plasmon excitation of SNB at about 650 nm is new and attract rather broad interest. But the discussion of the photocatalytic activity based on the plasmon excitation still contain several serious questions. Actually, the hydrogen evolution reaction with a low efficiency from an oxalic acid as a sacrificial reagent should not be called "water splitting" reaction. It is just a hydrogen evolution reaction with a sacrificial reagent.

So, I cannot recommend publication of the revised manuscript in Nature Communication. My suggestion is to focus more on the origin of the red color.

Reviewer #2 (Remarks to the Author):

The authors have worked hard to address all the concerns of the referees. In general, most of the issues were addressed. However, there is still one major error. The authors are assuming that plasmon mediated light absorption enables electronic transitions that typically are not allowed (the CB to B1 $d \rightarrow d$ transition in this case). This is not strictly true. These processes still must be facilitated either by phonons or geometry. The authors are advised to read a nice recent paper from Atwaters group on this (ACS Nano 10, 957-966), and include correct descriptions of how the forbidden transitions are excited through plasmon dephasing.

Reviewer #3 (Remarks to the Author):

The revised structural explanation and DFT analysis is helpful and convincing. In fact, this portion of the paper along with the Hall measurements of carrier properties is a compelling story, with the optical measurements contributing a secondary layer of narrative. A coherent combination of these observations would make this article compelling for the Nature Comm audience. However, the revised manuscript lacks clarity in the communication of some ideas that is misleading, as described in the comments below.

The reviewer response letter gives a lot more justification for a plasmon-mediated e-h generation process, and good evidence for why an interband transition is NOT directly excited. The updated description of the SPR excitation \rightarrow Landau damping \rightarrow B1 state e-h generation \rightarrow carrier cooling/e-h extraction is compelling. It would be helpful to include some of this narrative in the main text or SI to guide the reader's understanding as well. Furthermore, there isn't any actual discussion of the catalysis experiments (Figure S5) in the main text, although this is not a new result and it is reasonable to simply include the justification of the photocatalytic behavior of the material in the SI. A reader who isn't well versed in the literature of SNO may appreciate more description of this experiment, especially considering that a dark (no-illumination) control experiment wasn't included in Figure S5.

Explanation of the plasmon resonance is lacking, although this may be a lack of communication on the authors' part rather than a mistake in analysis. First of all, it is unclear how the UV-vis-NIR and ellipsometry data relate in terms of the fitted Drude model for each. Was the reflectivity data fitted in Figure 2b using the same optical parameters applied to the fitted ellipsometry data in Figure S7?

The reviewer response only describes adjustments to the fitting for the ellipsometry data, not the UV-vis-NIR reflectivity data. Also, the assertion that a bulk (volume) plasmon resonance is being excited is not supported by theory or literature. See (1) Xia, C.; Yin, C.; Kresin, V. V. Physical Review Letters 2009, 102, 156802 or (2) Ferrell, R. A. Phys. Rev. 1958, 111, 1214 for a discussion of the limitations of describing a volume optically-excited plasmon resonance in a metal. The ellipsometric analysis performed with an anisotropic Drude model is dependent on a thin film (~ 200 nm) surface stack, so a surface plasmon would be a reasonable explanation for the observed data. However, the actual results of the Drude fit, including calculated parameters such as free carrier concentration, background dielectric constant, mobility and geometry constraints should be included in the SI, or a justification for why they were omitted.

The description of the TA is well justified in terms of the decay lifetimes, but the DR spectra are explained in a speculative manner that is neither convincing nor helpful to the interpretation of their experiments. In fact, it clouds their otherwise coherent explanations of optical data. The DR measurements look at changes in reflectance, but the positive and negative peaks are loosely attributed to 'absorbance' from completely different features, without accounting for a population/depopulation of particular states that would cause a reflectance bleaching/darkening at either feature. The descriptions of these features should be clarified for the reader. The response to my earlier comments about the apparent peak shift from the derivative shape of this feature

wasn't adequate to dispel my skepticism of this assignment, based on my understanding of earlier work of TA analysis of plasmons in gold nanostructures, e.g. by Ahmadi et al (Ahmadi, T. S.; Logunov, S. L.; El-Sayed, M. A. J Phys Chem 1996, 100, 8053). In fact, thermalization of hot carriers would be expected to shift or broaden the SPR loss function, which is a much simpler explanation than two concurrent, long lifetime transitions simultaneously bleaching and darkening with identical dynamics.

A clearer, more thorough and less speculative analysis of the plasmonic behavior of these films would make this paper more compelling. The experimental observations are interesting but the communication of the analysis is confusing and possibly misleading.

Reviewer #4 (Remarks to the Author):

The authors have addressed the referee comments. I recommend publication.

Response*

Reviewer #1 (Remarks to the Author):

Comment 1.1

Fig. S6 is the band diagram of the SNO that was presented by the authors. **At first, the sign of the vertical axis (potential vs. NHE) should be reversed.** Then, l. 153-162 in the text, the authors mentioned the holes of this material drive the oxidation reaction, i.e. the oxidation reaction of oxalic acid (not water oxidation reaction). **Where is the redox potential (or the oxidation potential of oxalic acid) in Fig. S6?**

Response 1.1

We thank the referee for pointing this out and we have replaced the Fig. S6 with Fig. 5 in the revised manuscript. Fig. 5 shows not only reduction potentials of reactions such as H^+/H_2 and $CO_2/H_2C_2O_4$, but also the working mechanism of photocatalytic hydrogen evolution on Pt loaded $SrNbO_3$. Please see the revised manuscript (p.13: all, p.14: l.1-4 and Fig.5) for more detail. It should be noted that the reduction potential of H^+/H_2 is larger than that of $CO_2/H_2C_2O_4$ which might mean the standard $CO_2/H_2C_2O_4$ electrode can be oxidized by the standard hydrogen electrode in a galvanic cell. However, as the oxalic acid is widely used as hole scavenger (reported in numerous papers on photocatalysis^{1, 2, 3, 4}), the reaction rate may be too low to be measured in

* To our understanding, we summarize the reviewers' comments in black. The issues or questions raised by the reviewers are highlighted in bold. Our response to them are in blue and modification made in manuscript are highlighted in yellow.

practice which is possible due to the relatively small potential difference of -0.43 V. We thank the referee for pointing out the sign of the potential which has been rectified.

Comment 1.2

From l. 179 to l. 211 in the text, the authors attempted to explain the mechanism of photocatalytic reaction based on the results of time resolved pump-probe measurements. **In the explanation of the results, the authors suddenly (?) introduced “deep trapped states”, which confused me to understand the mechanism. Actually, where are the deep trapped states in the diagram of Fig. S6?** In addition, I afraid the authors made a wrong interpretation about Fig. 4(d). The authors mentioned that plasmon resonance increases the lifetime of hot carriers because the lifetime of the transition from deep trapped states to the conduction band plotted in Fig. 4(d) drastically increases when the energy of pump pulse is higher than that of the plasmon resonance (Lines 198-211). **However, the authors also mentioned that the energy of transition from deep trapped states to conduction band and that of plasmonic resonance is approximately same (Lines 188-191). Therefore, the pump pulse with energy lower than plasmon resonance cannot excite an electron in deep trapped states to conduction band and the transient absorption spectroscopy in this energy region might probe other minor processes.**

Response 1.2

Thanks to Referee 3 who pointed us to the pump probe spectrum of gold nano particles where a similar spectrum was seen arising from plasmons, we have modified the interpretation in the revised manuscript (p.10: l.19-21, p.11-12: all). Please also

read the detailed interpretation for the assignment of DR peaks in the supplementary information (SI, p.14-17, "Interpretation for differential reflection (DR) peaks").

In brief, it is known that a short laser pulse can selectively heat the electrons in metals⁵. When the temperature of electrons increases, the intensity of plasmon band will decrease and its linewidth will increase⁶. For example, Fig. R1 shows the transient absorption spectra of gold nanoparticle⁵. One main peak is near the plasmon wavelength and two derivative peaks are at the wings of the main peak (where the downward arrows point in Fig. R1). The main peak is due to the reduced intensity of the "excited"-state plasmon band. The two wings are due to the broadening of plasmon band. Similar to the case in gold nanoparticle, the peak near 670 nm can be attributed to the decrease of the plasmon band intensity. The peak near 600 nm is a derivative peak due to the plasmon band broadening induced by the increased electron temperature. The other wing at the lower energies is not seen due to limitations of our spectrometer. In this revised interpretation, the defect states are excluded. Because both the 600 nm peak and 670 nm peak are related to thermal effect of conduction electrons which can be heated by the short laser pulse, the DR spectra with the lower energy pump pulse can be similar with those with higher energy pump pulse.

Fig. R1 | “Bleach” recovery of the ground state and the transient absorption spectra of the sample as a function of the time delay between the pump (600 nm) and probe (white light) laser beams. The inset show the kinetics of the plasmon band using 600 nm excitation light⁵.

Comment 1.3

From l. 212 to l. 225, the authors discussed band gap structures of three different compositions by DFT calculations in Fig. 5, i.e. SrNbO₃ (b), SrNbO_{3.4} (d) and SrNbO_{3.5} (f). **To the reviewer’s understanding from the text, the authors regarded SrNbO₃ (b) being corresponded to the sample prepared at the lowest O₂ pressure (5x10⁻⁶ Torr), which was mainly discussed in the present manuscript.** If it is correct, the band structure is the same one which was reported in the previous work of Nat. Mater. in 2012. Of course, it is quite reasonable to interpret the red colour of this material by plasmon resonance as proved by the former half of the present work. I believe that part is a nice piece of work, **but it is not reasonably connected to the photocatalytic reaction mechanism.**

Response 1.3

Yes, it's true that we regard SrNbO₃ as the sample prepared at the lowest O₂ pressure. This has been confirmed by a 3.04 MeV Oxygen resonance Rutherford backscattering experiment (Fig. S4).

In previous report of Nature Material in 2012, SNO is regarded as a visible light photocatalyst and the visible light absorption is considered due to the interband transition (CB to B₁)⁴. In this manuscript, SNO is also proved to be a visible light photocatalyst (Fig. S12). However, it should be noted that the visible light absorption is due to the plasmon resonance other than the interband transition (Fig. 2(b), (d)). In general, the first step of photocatalysis is photon absorption and thus directly decides the performance of a photocatalyst. In our manuscript, we first proved the existence of plasmon resonance in SNO. SNO is in fact an ~ 4 eV degenerate semiconductor with a large carrier density. Because the plasmon resonance is the only absorption peak in the visible range, interband transitions do not play a role. Our model of SNO is more accurate and it can help us to improve the performance of SNO as a photocatalyst.

Comment 1.4

In Fig. S5, the authors made an effort to show the photocatalytic activity of the SNB. Unfortunately, the activity of hydrogen evolution is very low in an aqueous solution of oxalic acid. **The authors should estimate the quantum efficiency of the reaction. In addition, what will be the maximum quantum efficiency of hot electron transfer to Pt from plasmon excitation, which would be an interesting question?**

Response 1.4

Thanks for this suggestion and we have performed the measurement of apparent quantum efficiency (AQE) at the range from 420 nm to 600 nm of SrNbO₃ (1 wt.% Pt) for photocatalytic H₂ evolution. The width of the monochromatic light used as the light source is ± 5 nm. The AQE at each wavelength was calculated from the ratio of twice the number of H₂ molecules to the number of incident photons as shown in equation R1:

$$AQE = \frac{2 \times \text{the number of } H_2 \text{ molecules}}{\text{the number of incident photons}} \times 100\% \quad (\text{Eq. R1})$$

Fig. R2 shows the AQE spectrum together with the optical absorbance spectrum of SrNbO₃. As we can see, the AQE start to increase from 500 nm to 600 nm and while the AQE values at the range from 420 nm to 480 nm are below the measurable limit. However, the light source used for the photocatalytic hydrogen evolution measurement on SrNbO₃ is in the visible range (>410 nm) and under its irradiation a significant hydrogen evolution rate can be obtained (Fig. S12). Thus we can deduce that the AQE will be relatively large at the range from 600 nm to 800 nm. This deduction is also in agreement with the absorbance spectrum of SrNbO₃. This paper is not focused on optimizing the Faraday efficiency but more towards understanding the mechanism. By ball milling and enhancing effective surface area, optimizing the Pt mixture and the optical radiation significant improvements to the AQE is possible but that is not the main focus of this paper.

Fig. R2 | The absorbance spectrum (Blue) and apparent quantum efficiency (AQE) spectrum (Brown) from 420 nm to 600 nm for photocatalytic hydrogen evolution of SrNbO₃. 1 wt. % of Pt is used as co-catalyst.

Comment 1.5

It is better to cite and to discuss the literature below:

Oka, D.; Hirose, Y.; Nakao, S.; Fukumura, T.; Hasegawa, T. Phys. Rev. B 2015, 92 , 205102.

Response 1.5

Thank you very much for providing a report of a very interesting work! This paper also reported the high carrier density of SrNbO₃ at room temperature. But the mobility of this SrNbO₃ thin film grown on KaTiO₃ substrate is one order larger than that of SrNbO₃ thin film grown on LaAlO₃ substrate. This should be mainly due to the much smaller

lattice mismatch between SrNbO_3 and KATiO_3 (mismatch = -0.85%) compared with that (mismatch = -5.77%) between SrNbO_3 and LaAlO_3 . We have included a discussion of this work in our revised paper (Highlighted in p.4: l.4-9, p.7: l.4-9).

Comment 1.6

As a whole, I agree the finding of plasmon excitation of SNB at about 650 nm is new and attract rather broad interest. But the discussion of the photocatalytic activity based on the plasmon excitation still contain several serious questions. Actually, the hydrogen evolution reaction with a low efficiency from an oxalic acid as a sacrificial reagent should not be called “water splitting” reaction. It is just a hydrogen evolution reaction with a sacrificial reagent.

Response 1.6

Based on previous report and our results, it should only be called “photocatalytic hydrogen evolution reaction”. We thank the referee for pointing this out.

It is a prevailing practice to measure the hydrogen evolution reaction with a sacrificial reagent for some novel photocatalyst in its preliminary stage. In this paper, our proposal that these plasmon-induced hot carriers generated from the decay of plasmons produced by absorption of sunlight should be responsible for the photocatalytic activity of this material is totally different from the interband transition model of Xiaoxiang Xu, et al⁴. SrNbO_3 may be the first plasmonic metallic oxide that could be used as photocatalyst. Though the photocatalytic efficiency of SrNbO_3 is not high at this stage, what we have learnt here could enable us to design better photocatalysts.

Comment 1.7

My suggestion is to focus more on the origin of the red color.

Response 1.7

Please see the above discussion in response 1.3.

Reviewer #2 (Remarks to the Author):

Comment 2.1

The authors have worked hard to address all the concerns of the referees. In general, most of the issues were addressed. However, there is still one major error. The authors are assuming that plasmon mediated light absorption enables electronic transitions that typically are not allowed (the CB to B1 $d \rightarrow d$ transition in this case). This is not strictly true. These processes still must be facilitated either by phonons or geometry. The authors are advised to read a nice recent paper from Atwaters group on this (ACS Nano 10, 957-966), and include correct descriptions of how the forbidden transitions are excited through plasmon dephasing.

Response 2.1

Based on the reference pointed to us by Referee 3, we have completely modified our explanation which is in excellent agreement with our data which does not have the above discrepancy. Please see the revised manuscript, main text (p.10: l.19-21, p.11-12: all) and supplementary (SI, p.14-17, "Interpretation for differential reflection (DR peaks") for more detail. Regarding the plasmon dephasing effect and the catalysis we

have addressed Atwater's publication in our revised discussion (See p.13: all and p.14: l.1-4 and Fig.5).

Reviewer #3 (Remarks to the Author):

The revised structural explanation and DFT analysis is helpful and convincing. In fact, this portion of the paper along with the Hall measurements of carrier properties is a compelling story, with the optical measurements contributing a secondary layer of narrative. A coherent combination of these observations would make this article compelling for the Nature Comm audience. However, the revised manuscript lacks clarity in the communication of some ideas that is misleading, as described in the comments below.

Comment 3.1

The reviewer response letter gives a lot more justification for a plasmon-mediated e-h generation process, and good evidence for why an interband transition is NOT directly excited. The updated description of the SPR excitation -> Landau damping -> B1 state e-h generation -> carrier cooling/e-h extraction is compelling. **It would be helpful to include some of this narrative in the main text or SI to guide the reader's understanding as well.**

Response 3.1

We have found your comments extremely helpful in better interpreting our data and so we are very grateful to you for this.

We have added one paragraph and Fig. 5 to clearly interpret the working mechanism of photocatalytic hydrogen evolution on Pt loaded SrNbO₃ in which the plasmon-induced electron-hole pair generation process is also interpreted in detail. Please see the revised manuscript (See p.13: all and p.14: l.1-4 and Fig.5).

Comment 3.2

Furthermore, there isn't any actual discussion of the catalysis experiments (Figure S5) in the main text, although this is not a new result and **it is reasonable to simply include the justification of the photocatalytic behavior of the material in the SI**. A reader who isn't well versed in the literature of SNO may appreciate more description of this experiment, **especially considering that a dark (no-illumination) control experiment wasn't included in Figure S5**.

Response 3.2

Please see the revised SI (p.18-19 and Fig. S12) for the justification of the photocatalytic behavior of SrNbO₃. The dark control experiment result and its justification are also shown in Fig.S12.

Comment 3.3

Explanation of the plasmon resonance is lacking, although this may be a lack of communication on the authors' part rather than a mistake in analysis. **First of all, it is unclear how the UV-vis-NIR and ellipsometry data relate in terms of the fitted Drude model for each. Was the reflectivity data fitted in Figure 2b using the same optical parameters applied to the fitted ellipsometry data in Figure S7?** The reviewer

response only describes adjustments to the fitting for the ellipsometry data, not the UV-vis-NIR reflectivity data.

Response 3.3

Here we use two different model to fit the UV-Vis-NIR reflection spectrum and ellipsometry data.

The UV-vis-NIR reflection spectrum was fitted by the widely used Drude-Lorentz model. The software of RefFit was used to perform the fitting and extract the plasmon frequency and the complex dielectric function spectrum (Fig.R3).

As we all know, the normal-incidence reflectivity can be expressed by the complex dielectric function $\varepsilon(\omega) = \varepsilon_1(\omega) + i\varepsilon_2(\omega)$ according to the Fresnel equation:

$$R(\omega) = \left| \frac{1 - \sqrt{\varepsilon(\omega)}}{1 + \sqrt{\varepsilon(\omega)}} \right|^2. \quad (\text{Eq.R2})$$

With the Drude-Lorentz dielectric function, the $\varepsilon(\omega)$ in the above equation is

$$\varepsilon(\omega) = \varepsilon_\infty + \sum_i \frac{\omega_{pi}^2}{\omega_{oi}^2 - \omega^2 - i\gamma_i\omega} \quad (\text{Eq.R3})$$

Where ε_∞ is the so called 'high-frequency dielectric constant', which represents the contribution of all oscillators at very high frequencies, the parameters ω_{pi} , ω_{oi} and γ_i are the plasmon frequency, the oscillator resonance frequency and the linewidth (scattering rate) respectively of the i-th Lorentz oscillator. For the Drude term, which describes to response of the unbound (free) charge carriers, the ω_{oi} is zero. In general, the real ($\varepsilon_1(\omega)$) and imaginary ($\varepsilon_2(\omega)$) parts of dielectric function are not independent and they are coupled by the Kramers-Kronig (KK) relation.

The Drude model used in the ellipsometry spectroscopy fitting uses carrier density and mobility as the parameters. For Drude tail, the oscillator frequency is 0 ($\omega_0=0$). The ω_p can be converted to the carrier density (n) because the plasmon frequency is dependent on the carrier density. The linewidth (γ) is proportional to $1/t$, where t is the mean scattering time ($1/t$ is the scattering rate), and carrier mobility (μ) is proportional to mean scattering time (t), thus γ is inversely proportional to μ . Therefore, the Drude-Lorentz model is equivalent to the model used in ellipsometry from the view of their physical essence though the fitting processes are different.

Fig.R3 | Complex dielectric function and loss function spectra of SrNbO₃ calculated by the fitting results of the film's reflection spectrum. The top spectrum shows the real part ($\epsilon_1(\omega)$) of complex dielectric function ($\epsilon(\omega) = \epsilon_1(\omega) + i\epsilon_2(\omega)$). Vertical dashed line indicates the zero-crossing of $\epsilon_1(\omega)$ of SrNbO₃. The bottom two spectra show the imaginary part ($\epsilon_2(\omega)$) of complex dielectric function (yellow line) and loss function ($-\text{Im}[\epsilon^{-1}(\omega)]$, blue line), respectively.

The fitted reflection spectrum beyond 600 nm is shown in Fig. 2(b). The real part ($\varepsilon_1(\omega)$) and imaginary ($\varepsilon_2(\omega)$) parts of dielectric function as well as the loss function ($-Im[\varepsilon^{-1}(\omega)]$) are shown in Fig. R3. The $\varepsilon_2(\omega)$ shows an intraband Drude feature at the energy range (<2.0 eV) which is due to the absorption of free carriers. The loss function shows a peak at 1.65 eV (~750 nm) which indicates the plasmon energy position. The $\varepsilon_1(\omega)$ crosses zero at 1.6 eV (~775 nm) which further confirms the energy position of plasmon.

The plasmon energy position obtained by the UV-Vis-NIR spectroscopy is smaller than the value obtained from ellipsometry spectroscopy because the UV-Vis-NIR measurement is a relatively rough method compared with the ellipsometry spectroscopy. The light source of UV-Vis-NIR spectroscopy is un-polarized and it can only measure the amplitude of light intensity while the ellipsometry spectroscopy uses the s-polarized and p-polarized light waves and it can measure both the amplitude ratio (ψ) and phase difference (Δ) between two different polarized light waves. Thus, the ellipsometry can obtain much more accurate optical information. The reflectance of substrate is also ignored in this Drude-Lorentz model because the LaAlO₃ substrate (Bandgap is 5.6 eV) is almost transparent for the light beyond 250 nm. However, this may induce some small error in the simulation result while in the model of ellipsometry we used two-layer structure (film and substrate) to simulate the result (SI, p7-12). Therefore, we think that on one hand the energy position obtained from the ellipsometry data is more accurate than that from UV-Vis-NIR data, on the other hand the similar fitting results between the UV-Vis-NIR data and ellipsometry data doubly confirm the plasmon energy position in strontium niobate.

Comment 3.4

Also, the assertion that a bulk (volume) plasmon resonance is being excited is not supported by theory or literature. See (1) Xia, C.; Yin, C.; Kresin, V. V. Physical Review Letters 2009, 102, 156802 or (2) Ferrell, R. A. Phys. Rev. 1958, 111, 1214 for a discussion of the limitations of describing a volume optically-excited plasmon resonance in a metal. **The ellipsometric analysis performed with an anisotropic Drude model is dependent on a thin film (~200nm) surface stack, so a surface plasmon would be a reasonable explanation for the observed data. However, the actual results of the Drude fit, including calculated parameters such as free carrier concentration, background dielectric constant, mobility and geometry constraints should be included in the SI, or a justification for why they were omitted.**

Response 3.4

Thank you very much for pointing this out! As the two references reported, the volume plasmon cannot be excited by direct photo-absorption^{7, 8}. Thus, we agree that the plasmon excited in SrNbO₃ when under irradiation is likely to be the surface plasmon rather than the volume plasmon. Fig. S9 shows the ellipsometric data based calculation of the penetration depth versus photon energy. Coupling of photons to the SrNbO₃ plasmons is not an issue in the case of the powders used in the photocatalytic hydrogen evolution. The penetration depth at the plasmon resonance is about 80 nm which is a significant fraction of the nano particle diameter (~100-1000 nm). So to be sure of whether the plasmon is surface versus bulk we will need to do further experiments and this point has been addressed in the text (p.9: l.16-22, p.10: l.1-2).

The reflection differences shown in the DR spectra are the results of the high temperature carrier induced shifting and broadening of the plasmon absorption band (as will be discussed in response to 3.5 in detail). This is similar with the TA analysis of plasmons in gold nanoparticles reported by Ahmadi et al ⁵ as mentioned in comment 3.5. The absorption peak observed in the UV-Vis-NIR absorption spectrum of SrNbO₃ (Fig.2(b)) should also be attributed to the plasmon absorption as there is no mid-gap state between the conduction band and B₋₁ band as shown in Fig. S13 (new data acquired recently). This new interpretation not only can be used to better understand the DR spectra, but also addresses the concerns in the comment 1.2. As we discussed in the detailed analysis of the DR peaks in supplementary information (SI, p.14-17, “Interpretation for differential reflection (DR) peaks”), the high temperature carrier induced plasmon shifting and broadening is the only possible reason to explain all the interesting phenomena we observed, especially the similar dynamics of the observed two peaks (600 nm and 670 nm) in DR spectra and why the DR spectra are almost independent of excitation wavelength. Thus, it can be concluded that the plasmon exists in SrNbO₃ and it can be excited near 670 nm when SrNbO₃ is under light irradiation. The ellipsometry spectrum can measure the dielectric function and the energy position of the plasmon via the loss function.

The fitting parameters of the carrier density and mobility have been added to the Table.S2 in the supplementary information (Highlighted in p.9 of SI).

Comment 3.5

The description of the TA is well justified in terms of the decay lifetimes, but the DR spectra are explained in a speculative manner that is neither convincing nor helpful to

the interpretation of their experiments. In fact, it clouds their otherwise coherent explanations of optical data. **The DR measurements look at changes in reflectance, but the positive and negative peaks are loosely attributed to ‘absorbance’ from completely different features, without accounting for a population/depopulation of particular states that would cause a reflectance bleaching/darkening at either feature. The descriptions of these features should be clarified for the reader.** The response to my earlier comments about the apparent peak shift from the derivative shape of this feature wasn’t adequate to dispel my skepticism of this assignment, based on my understanding of earlier work of TA analysis of plasmons in gold nanostructures, e.g. by Ahmadi et al (Ahmadi, T. S.; Logunov, S. L.; El-Sayed, M. A. J Phys Chem 1996, 100, 8053). **In fact, thermalization of hot carriers would be expected to shift or broaden the SPR loss function, which is a much simpler explanation than two concurrent, long lifetime transitions simultaneously bleaching and darkening with identical dynamics.**

Response 3.5

This reference made all the difference to us! Thank you for pointing this out.

We totally agree that high temperature carriers shifting and broadening the plasmon loss function is a better explanation for our DR spectra and addresses the concerns of Referee 1 as well (comment 1.2).

We have incorporated this new explanation in our revised manuscript (p.10: l.19-21, p.11-12: all) and added a new section to explain the assignment of the DR peaks in detail in the supplementary information (SI, p.14-17, “Interpretation for differential reflection (DR) peaks”).

Reviewer #4 (Remarks to the Author):

Comment 4.1

The authors have addressed the referee comments. I recommend publication.

Response 4.1

We would like to thank Reviewer #4 for his/her support for our manuscript!

Reference

1. Kmetykó, Á. *et al.* Photocatalytic H₂ Production Using Pt-TiO₂ in the Presence of Oxalic Acid: Influence of the Noble Metal Size and the Carrier Gas Flow Rate. *Materials* **7**(10), 7022-7038 (2014).
2. Iliev, V. *et al.* Photocatalytic properties of TiO₂ modified with gold nanoparticles in the degradation of oxalic acid in aqueous solution. *Appl. Catal., A* **313**(2), 115-121 (2006).
3. Mogyorósi, K. *et al.* Comparison of the substrate dependent performance of Pt-, Au- and Ag-doped TiO₂ photocatalysts in H₂-production and in decomposition of various organics. *React. Kinet. Catal. Lett.* **98**(2), 215-225 (2009).
4. Xu, X. Randorn, C. Efstathiou, P. & Irvine, J. T. S. A red metallic oxide photocatalyst. *Nat. Mater.* **11**, 595-598 (2012).
5. Ahmadi, T. S. Logunov, S. L. & El-Sayed, M. A. Picosecond Dynamics of Colloidal Gold Nanoparticles. *J. Phys. Chem.* **100**(20), 8053-8056 (1996).
6. Doremus, R. H. Optical Properties of Small Gold Particles. *J. Phys. Chem.* **40**(8), 2389 (1964).
7. Myung, C. W. *et al.* Finite Amplitude Effects on Landau Damping and Diminished Transportation of Trapped Electrons. *J. Phys. Soc. Jpn.* **83**(7), 074502 (2014).
8. Xia, C. Yin, C. & Kresin, V. V. Photoabsorption by volume plasmons in metal nanoclusters. *Phys Rev Lett* **102**(15), 156802 (2009).

REVIEWERS' COMMENTS:

Reviewer #1 (Remarks to the Author):

This manuscript has become much better than the previous version. Especially, the interpretation of the time-resolved spectra seems to be quite reasonable and it gives a novel information to the broad readers. Concerning the reaction mechanism depicted in Figure 5, I still have some doubts, especially the reaction route of excited holes. However, I understand it is difficult to fully reveal the reaction mechanism at present stage and it is probably beyond the scope of this manuscript.

One of the additional comments is as follows:

In Supplementary Information, the authors discussed on the activity of Pt/SrNbO₃ for hydrogen evolution based on the unit of $\mu\text{mol}/(\text{g}\cdot\text{h})$. Although this kind of normalization as well as $\mu\text{mol}/(\text{m}^2\cdot\text{h})$ using a specific surface area is useful and reasonable for normal/thermal heterogeneous catalysis reactions, it is misleading to express the activity or efficiency of photocatalysis. The proper way is to show the quantum efficiencies at specific wavelengths. If it is not available, it is more accurate to show the activity in $\mu\text{mol}/\text{h}$ with indicating the "real" amounts of used catalyst and volume of the solution, mg/mL with a light intensity of some wavelength region or the kind of light source. In almost all cases, photocatalytic activities do not increase linearly with the amounts of photocatalysts in the same volumes of solution and a light source. As a whole, I now recommend publication of this manuscript in Nat. Commun.

Reviewer #2 (Remarks to the Author):

All of my issues have been addressed.

Reviewer #3 (Remarks to the Author):

The authors have done a commendable job addressing the referees' comments. I have one comment regarding the TA measurements: a closer inspection of the TA data as it is presented in Fig. 4(A,B), Fig S10(A,B) and Fig S11(A) seems to reveal confusing non-monotonic behavior in the $<100\text{ps}$ range however, or an inconsistency in normalization between the fast and slow processes measured in the manuscript. Within the first 2.0 ps (Fig S11A) the intensity drops by about half for all wavelengths, with a negative slope tail. However, in Fig 4B there appears to be an increase and subsequent decrease around 25ps, corresponding to a change of about 10-20% for each wavelength. Thus, the DR data plotted in Fig S10A and Fig 4A appear to lie on either side of the spike in intensity measured at $\sim 25\text{ps}$ in Fig 4b. This appears to account for the similar absolute values of the DR spectra at 5ps and 1000ps, but it is confusing when described in terms of the decay processes of the narrative. This may just be an artifact of the measurement, data presentation or normalization, but it may confuse the reader. I don't think this issue is a significant barrier to publication of the manuscript, but the authors should add an explanation for the benefit of the reader. Regardless of this one issue, I recommend publication.

Response to Referees*

Reviewer #1 (Remarks to the Author):

Comment 1.1

This manuscript has become much better than the previous version. Especially, the interpretation of the time-resolved spectra seems to be quite reasonable and it gives a novel information to the broad readers. **Concerning the reaction mechanism depicted in Figure 5, I still have some doubts, especially the reaction route of excited holes.** However, I understand it is difficult to fully reveal the reaction mechanism at present stage and it is probably beyond the scope of this manuscript.

Response 1.1

We thank the referee for appreciating the improvements to the manuscript. In metallic thin films, it is known that electron hole pairs are generated by plasmons through Landau damping involving both intra and inter band transitions. This has been studied in detail in the case of gold, silver and aluminum showing the variable roles of intra versus inter band transition. Unfortunately, our understanding of the band structure of SrNbO₃ is still incomplete but for DFT calculations. We interpret the hot holes to be excited during the decay of plasmon via Landau damping and believe that the hot holes left in the electron deficient SrNbO₃ would have enough energy to induce the oxidation

* To our understanding, we summarize the reviewers' comments in black and the issues or questions raised by the reviewers are highlighted in bold. Our responses to them are in blue and modifications made in the manuscript are highlighted in yellow.

reaction with oxalic acid. Much as the referee pointed out experimental measurement of the band structure will be necessary to quantify the roles of specific bands and particularly, intra versus inter band transitions. We are planning such study for the future.

Comment 1.2

One of the additional comments is as follows:

In Supplementary Information, the authors discussed on the activity of Pt/SrNbO₃ for hydrogen evolution based on the unit of $\mu\text{mol}/(\text{g}\cdot\text{h})$. Although this kind of normalization as well as $\mu\text{mol}/(\text{m}^2\cdot\text{h})$ using a specific surface area is useful and reasonable for normal/thermal heterogeneous catalysis reactions, it is misleading to express the activity or efficiency of photocatalysis. **The proper way is to show the quantum efficiencies at specific wavelengths. If it is not available, it is more accurate to show the activity in $\mu\text{mol}/\text{h}$ with indicating the “real” amounts of used catalyst and volume of the solution, mg/mL with a light intensity of some wavelength region or the kind of light source.** In almost all cases, photocatalytic activities do not increase linearly with the amounts of photocatalysts in the same volumes of solution and a light source.

Response 1.2

We thank the reviewer for pointing this out. We have modified the related parts in the supplementary information and Supplementary Figure 12. However, the previous interpretation was also included to compare our efficiency with the reported one which

used the unit of $\mu\text{mol}/(\text{g}\cdot\text{h})$. Please see the revised Supplementary Information (Supplementary Note 1 (p.17) and Supplementary Figure 12) for more detail.

Comment 1.3

As a whole, I now recommend publication of this manuscript in Nat. Commun.

Response 1.3

We would like to thank Reviewer #1 for his/her support for our manuscript! We would like to acknowledge the benefit of the comments of the referee in helping us improve the manuscript.

Reviewer #2 (Remarks to the Author):

Comment 2.1

All of my issues have been addressed.

Response 2.1

We would like to thank Reviewer #2 for his/her support for our manuscript and the incisive comments!

Reviewer #3 (Remarks to the Author):

Comment 3.1

The authors have done a commendable job addressing the referees' comments. I have one comment regarding the TA measurements: a closer inspection of the TA data as it is presented in Fig. 4(A,B), Fig S10(A,B) and Fig S11(A) seems to reveal confusing non-monotonic behavior in the <100ps range however, or an inconsistency in normalization between the fast and slow processes measured in the manuscript. Within the first 2.0 ps (Fig S11A) the intensity drops by about half for all wavelengths, with a negative slope tail. **However, in Fig 4B there appears to be an increase and subsequent decrease around 25ps, corresponding to a change of about 10-20% for each wavelength. Thus, the DR data plotted in Fig S10A and Fig 4A appear to lie on either side of the spike in intensity measured at ~25ps in Fig 4b. This appears to account for the similar absolute values of the DR spectra at 5ps and 1000ps, but it is confusing when described in terms of the decay processes of the narrative.** This may just be an artifact of the measurement, data presentation or normalization, but it may confuse the reader. I don't think this issue is a significant barrier to publication of the manuscript, but the authors should add an explanation for the benefit of the reader. Regardless of this one issue, I recommend publication.

Response 3.1

Foremost, we would like to express our special thanks to Reviewer #3 for his/her support for our manuscript! We are extremely grateful to the referee for bringing to our attention the valuable reference of Ahmadi et al which enabled us to interpret our data in a consistent way.

The issue of the spike near ~25 ps should be attributed to measurement artifact relating to our optical setup. We have included a short explanation of this in the revised Supplementary Information (Supplementary Note 3, p.23)

Reference

1. Zhu, Y. T. Dai, Y. Lai, K. R. Li, Z. J. & Huang, B. B. Optical Transition and Photocatalytic Performance of d(1) Metallic Perovskites. *J. Phys. Chem. C* **117**(11), 5593-5598 (2013).
2. Xu, X. Randorn, C. Efstathiou, P. & Irvine, J. T. S. A red metallic oxide photocatalyst. *Nat. Mater.* **11**, 595-598 (2012).